# GENERATIVE REPRESENTATIONAL INSTRUCTION TUNING

**Niklas Muennighoff** [c]    **Hongjin Su** [h]    **Liang Wang** [m]    **Nan Yang** [m]

**Furu Wei** [m]  **Tao Yu** [h]  **Amanpreet Singh** [c]  **Douwe Kiela** [c]

[c] Contextual AI    [h] The University of Hong Kong    [m] Microsoft Corporation

niklas@contextual.ai

## ABSTRACT

All text-based language problems can be reduced to either generation or embedding. Current models only perform well at one or the other. We introduce generative representational instruction tuning (GRIT) whereby a large language model is trained to handle both generative and embedding tasks by distinguishing between them through instructions. Compared to other open models, our resulting GRITLM 7B is among the top models on the Massive Text Embedding Benchmark (MTEB) and outperforms various models up to its size on a range of generative tasks. By scaling up further, GRITLM 8x7B achieves even stronger generative performance while still being among the best embedding models. Notably, we find that GRIT matches training on only generative or embedding data, thus we can unify both at no performance loss. Among other benefits, the unification via GRIT speeds up Retrieval-Augmented Generation (RAG) by >60% for long documents, by no longer requiring separate retrieval and generation models. Models, code, etc. are freely available at https://github.com/ContextualAI/gritlm.

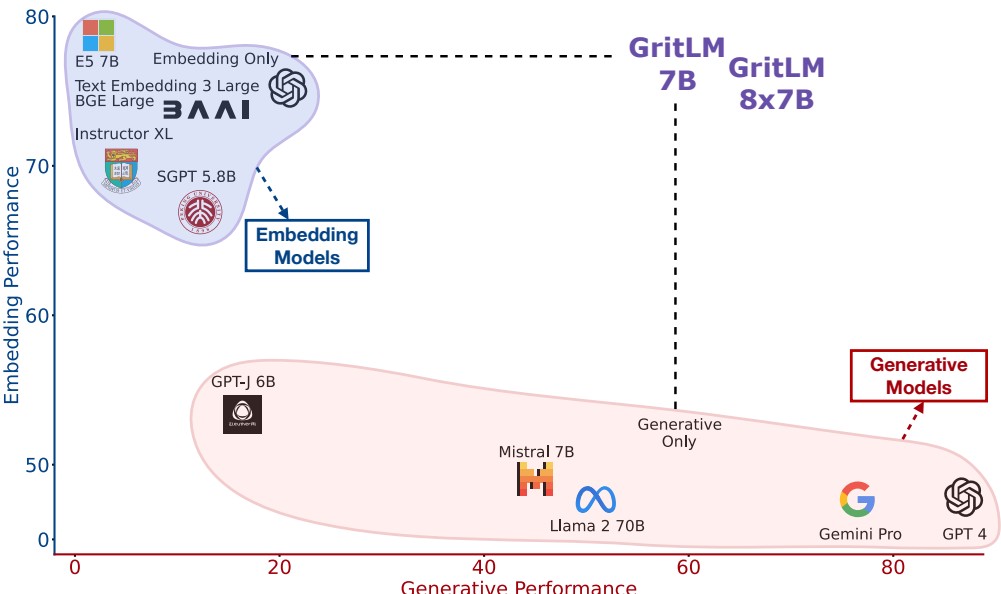

Figure 1: **Performance of various models on text representation (embedding) and generation tasks.** GRITLM is the first model to perform strongly at both types of tasks simultaneously.

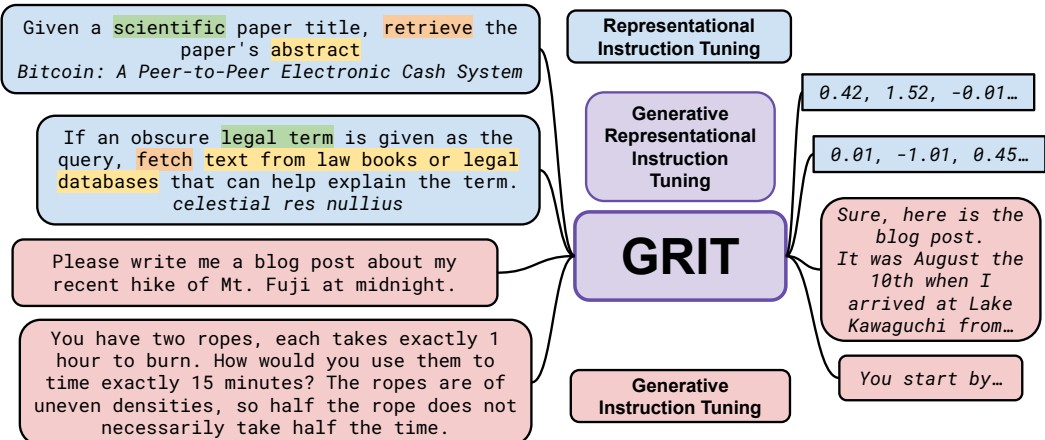

Figure 2: **GRIT.** The same model handles both text representation and generation tasks based on a given instruction. For representation tasks, instructions ideally contain target domain , intent , and unit (Asai et al., 2022). The representation is a numeric tensor, while the generative output is text.

# 1 INTRODUCTION

Creating a single general model that performs well at a wide range of tasks has been a long-standing goal of the field of artificial intelligence (Kaiser et al., 2017; Jaegle et al., 2021; Cho et al., 2021; Reed et al., 2022; Singh et al., 2022). Recently, large language models (LLMs) have emerged as a promising direction for a single multi-task model (Radford et al., 2019; Brown et al., 2020). Prior work has argued that all text-based language problems can be reduced to generation and thus handled by a single LLM (Raffel et al., 2023; Du et al., 2021).

However, tasks that use embeddings, such as clustering or retrieval (Muennighoff et al., 2023c), have largely been ignored from this perspective. Today, text embeddings power many real-world applications ranging from search engines to user-facing chatbots (Huang et al., 2020; Su et al., 2017). While integrating text embeddings into the generative paradigm is possible by generating a sequence of numbers to form the embedding tensor, it becomes impractical due to the high dimensionality and precision requirements of embeddings. Thus, it is more common and much easier to use the hidden state of the model as the embedding representation, which is already a numeric tensor (Muennighoff, 2022; Wang & Kuo, 2020; Morris et al., 2023). However, for current generative models this leads to poor performance. For example, while the T5 model (Raffel et al., 2023; Sanh et al., 2022) can handle any generative task in a sequence-to-sequence fashion, it requires finetuning to make its hidden state useful for text embedding (Ni et al., 2021a;b) during which it loses its generative capabilities.

We introduce GRIT (generative representational instruction tuning) which unifies embedding and generative tasks, leading to a model that excels at both tasks as shown in Figure 1. Figure 2 depicts how GRIT combines two previously disjoint training paradigms: (1) *Generative instruction tuning*, whereby the model is trained to respond to instructions by generating an answer (Wei et al., 2022; Sanh et al., 2022); and (2) *Representational instruction tuning*, whereby the model is trained to represent a provided input according to an instruction (Su et al., 2023; Asai et al., 2022). Via the instructions and separate loss functions the model learns to differentiate the two streams. We test our approach on models with up to 47B parameters. This unification via GRIT leads to three advantages:
**a) Performance:** Our unified model matches the performance of embedding-only and generative-only variants, even outperforming them on some tasks. At 7B parameters, GRITLM is among the best models on the Massive Text Embedding Benchmark (Muennighoff et al., 2023c) and at the same time outperforms some larger models on generative tasks, e.g., Llama 2 70B. By scaling further, GRITLM 8x7B has even better generative performance while only using 13B parameters at inference due to its MoE architecture (Jiang et al., 2024). Further, as our models use sliding window attention (Child et al., 2019; Beltagy et al., 2020), they can handle generative and embedding inputs of any length.
**b) Efficiency:** Generative and embedding models are often used together to make up for each other's

deficiencies (Guu et al., 2020; Lewis et al., 2021a). One such scenario is Retrieval-Augmented Generation (RAG) (Lewis et al., 2021a), where an embedding model retrieves context for a generative model to answer a user query. This requires passing the user query and the context into both the generative and the embedding model for a total of four forward passes. With GRITLM, the embedding and generative model are equivalent, allowing us to cache computations and halve the required number of forward passes. We find this can lead to $> 60\%$ faster RAG at inference with long documents.
**c) Simplicity:** Currently, API providers such as OpenAI provide separate generative and embedding endpoints. This requires separate load balancing, additional storage, and more complex serving software. A single model that handles both use cases significantly simplifies infrastructure needs.

Compared to generative instruction tuning, the main downside of GRIT is that it requires more finetuning compute due to training with two objectives. However, finetuning is generally cheap compared to pretraining, thus we think the benefits vastly outstrip this problem. Further, when considering training a separate generative and embedding model from scratch (e.g. for RAG), GritLM is generally cheaper when incorporating the pretraining compute, as there is only one pretraining and finetuning for GritLM, not separate ones for both a generative and an embedding model. Thus we recommend practitioners building instruction-following language models to adopt GRIT.

Alongside GRIT, we introduce novel performance improvements for embedding models including the use of bidirectional attention with mean pooling for LLM embeddings and making in-batch negatives stem from the same dataset rather than any dataset, as well as novelties for generative models such as mixing sample- and token-level loss aggregation. We ablate these in detail in Appendix B. We also propose new ways to reduce memory requirements during embedding model training in Appendix L.

## 2 GRIT

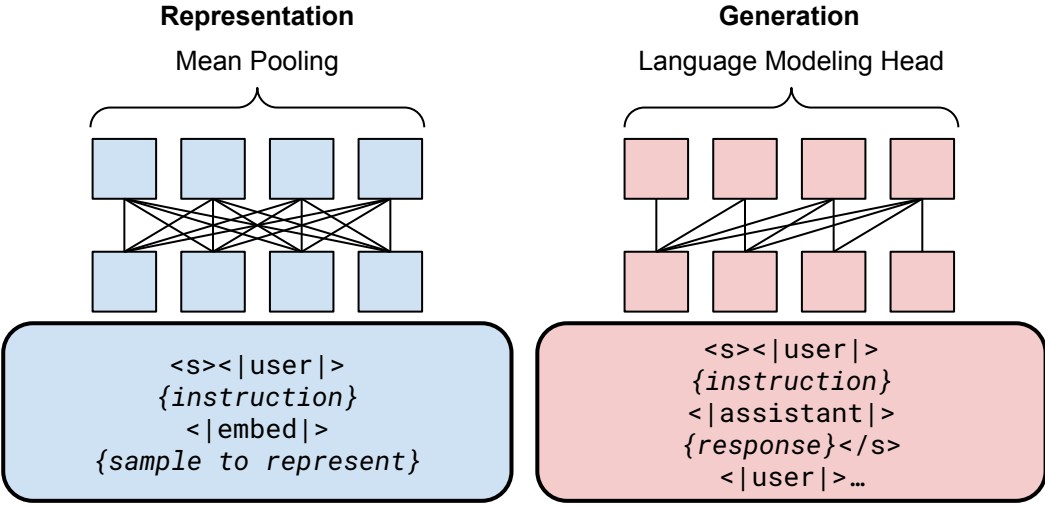

Figure 3: **GRITLM architecture and format.** *Left:* GRITLM uses bidirectional attention over the input for embedding tasks. Mean pooling is applied over the final hidden state to yield the final representation. *Right:* GRITLM uses causal attention over the input for generative tasks. A language modeling head on top of the hidden states predicts the next tokens. The format supports conversations with multiple turns (indicated with "...").

GRIT unifies representational instruction tuning (Su et al., 2023; Asai et al., 2022; Wang et al., 2024) and generative instruction tuning (Wei et al., 2022; Sanh et al., 2022; Muennighoff et al., 2023d) into a single model. We finetune a pretrained LLM (Brown et al., 2020) with embedding and generative instruction data in a consistent format (Figure 3). For embedding data, we follow prior work and use a contrastive objective with in-batch negatives (Chen et al., 2020; Gao et al., 2022):

$$\mathcal{L}_{\text{Rep}} = -\frac{1}{M} \sum_{i=1}^{M} \log \frac{\exp(\tau \cdot \sigma(f_\theta(q^{(i)}), f_\theta(d^{(i)})))}{\sum_{j=1}^{M} \exp(\tau \cdot \sigma(f_\theta(q^{(i)}), f_\theta(d^{(j)})))} \tag{1}$$

where $f$ is GRITLM parametrized by the model $\theta$, $\tau$ is a temperature hyperparameter and $\sigma$ corresponds to pooling applied to each output followed by cosine similarity. $q$ and $d$ are query and document samples. As depicted in Figure 3, we use bidirectional attention followed by mean pooling, which corresponds to averaging the hidden states across the sequence length. During pooling, we only average the final hidden states of the input sample, ignoring the instruction and format tokens. However, the instruction and format tokens still influence the final representation through the self-attention mechanism (Vaswani et al., 2023).

To compute the loss on generative data, we use the language modeling objective whereby the model needs to predict the next token (Radford et al., 2018; 2019):

$$\mathcal{L}_{\text{Gen}} = -\frac{1}{N} \sum_{i=1}^{N} \log P(f_{\theta,\eta}(x^{(i)})|f_{\theta,\eta}(x^{(<i)})) \tag{2}$$

where $f$ is GRITLM with parameters $\theta$ and language modeling head $\eta$, which is only used for generation. $x$ are generative training samples. We only compute loss over predicted tokens i.e. "`{response}`" in Figure 3. A key consideration is how to aggregate the generative loss. Aggregating at the sample level corresponds to giving each sample the same weight within a batch regardless of its token count. Such aggregation is commonly used for instruction tuning, as it can boost performance on discriminative tasks (Muennighoff et al., 2023d). However, Muennighoff et al. (2023d) also show how this in turn can lead to a model biased toward short generations. Meanwhile, aggregation at the token level corresponds to giving each token the same weight, thus samples with many tokens become more important. This leads to a model producing longer generations, which can be important for performance on generative tasks. Especially, human or machine-evaluated generative tasks, such as AlpacaEval (Li et al., 2023b), are known to be biased toward preferring longer generations (Wang et al., 2023). Note that when every sample has the same sequence length such as during pretraining or when the batch size is 1, token and sample level generative loss are equal to each other. One can mix the two to balance their trade-offs, for example doing token level loss across a subset of the batch and then giving each subset the same weight. We explore the trade-offs in our ablations in Appendix B. We sum the objectives with optional loss weights $\lambda_{\text{Rep}}$ and $\lambda_{\text{Gen}}$:

$$\mathcal{L}_{\text{GRIT}} = \lambda_{\text{Rep}}\mathcal{L}_{\text{Rep}} + \lambda_{\text{Gen}}\mathcal{L}_{\text{Gen}} \tag{3}$$

Notably, our formulation supports differing numbers of embedding samples ($M$) and generative samples/tokens ($N$). This allows for significantly increasing the embedding batch size while keeping the generative batch size fixed. A large embedding batch size is often key to well-performing text embedding models (Xiao et al., 2023) at the cost of requiring more compute at each step.

## 3 EXPERIMENTS

In this section, we first outline our experimental setup in §3.1. In §3.2, we discuss and benchmark the embedding and generative performance of our models. In Appendix B, we ablate the settings that led to our final models, including training data, precision, pooling, sequence length, and loss weights.

### 3.1 SETUP

We finetune our final models from Mistral 7B (Jiang et al., 2023) and Mixtral 8x7B (Jiang et al., 2024) using adaptations of E5 (Wang et al., 2024) and the Tülu 2 data (Ivison et al., 2023). For E5, we adapt it by adding S2ORC (Lo et al., 2020) to increase its scientific data ("E5S"), while for Tülu 2 we filter out their custom prompts that contain answers related to the origin of their model. For GRITLM 7B, we use a batch size of 2048 for embedding data and 256 for generative data and we train the model for a total of 1253 steps corresponding to one epoch on the generative data and 1.36 epochs on the embedding data. For GRITLM 8X7B, the embedding batch size is 256 due to compute limitations. We use several strategies to reduce the memory required during training including a novel technique to split the embedding triplet into separate forward and backward passes detailed in Appendix L. Other hyperparameters are detailed in the ablation experiments in Appendix B and Appendix M. For embedding performance we evaluate using the 56 main datasets from MTEB (Muennighoff et al., 2023c). For generative performance, we largely follow the evaluation setup of Ivison et al. (2023) except that we use the HumanEvalSynthesize (Muennighoff et al., 2023a) variant of HumanEval, as it is more adequate for instruction-following models. We explain each task in detail in Appendix I.

Table 1: **Embedding performance of GRITLM and others**. We indicate parameter counts where available (B=billions). See Appendix I for task, metric, and dataset details. Appendix K contains per-dataset results of GRITLM models. LLMs not finetuned for embedding (Llama 2 70B, Mistral 7B (Instruct), GPT-J 6B, Gen.-only) are evaluated with weighted-mean pooling (Muennighoff, 2022). ♥Results from the MTEB leaderboard (https://hf.co/spaces/mteb/leaderboard)

| Task (→) | CLF | Clust. | PairCLF | Rerank | Retrieval | STS | Summ. | Avg. |
|---|---|---|---|---|---|---|---|---|
| Metric (→) | Acc. | V-Meas. | AP | MAP | nDCG | Spear. | Spear. | |
| Dataset # (→) | 12 | 11 | 3 | 4 | 15 | 10 | 1 | 56 |
| Proprietary models♥ | | | | | | | | |
| OpenAI v3 | 75.5 | 49.0 | 85.7 | 59.2 | 55.4 | 81.7 | 29.9 | 64.6 |
| Other Open Models♥ | | | | | | | | |
| Llama 2 70B | 60.4 | 29.0 | 47.1 | 38.5 | 9.0 | 49.1 | 26.1 | 35.6 |
| Mistral 7B | 63.5 | 34.6 | 53.5 | 43.2 | 13.2 | 57.4 | 19.7 | 40.5 |
| Mistral 7B Instruct | 67.1 | 34.6 | 59.6 | 44.8 | 16.3 | 63.4 | 25.9 | 43.7 |
| GPT-J 6B | 66.2 | 39.0 | 60.6 | 48.9 | 19.8 | 60.9 | 26.3 | 45.2 |
| SGPT BE 5.8B | 68.1 | 40.3 | 82.0 | 56.6 | 50.3 | 78.1 | 31.5 | 58.9 |
| Instructor XL 1.5B | 73.1 | 44.7 | 86.6 | 57.3 | 49.3 | 83.1 | **32.3** | 61.8 |
| BGE Large 0.34B | 76.0 | 46.1 | 87.1 | 60.0 | 54.3 | 83.1 | 31.6 | 64.2 |
| E5 Mistral 7B | 78.5 | 50.3 | **88.3** | 60.2 | 56.9 | **84.6** | 31.4 | 66.6 |
| **GRITLM** | | | | | | | | |
| Gen.-only 7B | 65.4 | 32.7 | 54.2 | 43.0 | 13.7 | 60.2 | 21.1 | 41.2 |
| Emb.-only 7B | 78.8 | **51.1** | 87.1 | **60.7** | **57.5** | 83.8 | 30.2 | **66.8** |
| GRITLM 7B | **79.5** | 50.6 | 87.2 | 60.5 | 57.4 | 83.4 | 30.4 | **66.8** |
| GRITLM 8x7B | 78.5 | 50.1 | 85.0 | 59.8 | 55.1 | 83.3 | 29.8 | 65.7 |

Table 2: **Generative performance of GRITLM and others**. We indicate parameter counts where available (B=billions). See Appendix I for dataset, setup, and metric details. ♥Results from Ivison et al. (2023) except for numbers marked with ♦ which are from Touvron et al. (2023) and † which are from us. For models that cannot be easily used as chat models, we set Alpaca to 0.

| Dataset (→) | MMLU | GSM8K | BBH | TyDi QA | HumanEval | Alpaca | Avg. |
|---|---|---|---|---|---|---|---|
| Setup (→) | 0 FS | 8 FS, CoT | 3 FS, CoT | 1 FS, GP | 0 FS | 0 FS, 1.0 | |
| Metric (→) | EM | EM | EM | F1 | pass@1 | % Win | |
| Proprietary models♥ | | | | | | | |
| GPT-4-0613 | 81.4 | 95.0 | 89.1 | 65.2 | 86.6† | 91.2 | 84.8 |
| Other Open Models♥ | | | | | | | |
| GPT-J 6B | 27.7 | 2.5 | 30.2 | 9.4 | 9.8 | 0.0 | 13.3 |
| SGPT BE 5.8B | 24.4 | 1.0 | 0.0 | 22.8 | 0.0 | 0.0 | 8.0 |
| Zephyr 7B β | 58.6 | 28.0 | 44.9 | 23.7 | 28.5 | 85.8 | 44.9 |
| Llama 2 70B | 64.5 | 55.5 | 66.0 | **62.6** | 29.9♦ | 0.0 | 46.4 |
| Llama 2 Chat 13B | 53.2 | 9.0 | 40.3 | 32.1 | 19.6† | 91.4 | 40.9 |
| Llama 2 Chat 70B | 60.9 | 59.0 | 49.0 | 44.4 | 34.3† | 94.5 | 57.0 |
| Tülu 2 7B | 50.4 | 34.0 | 48.5 | 46.4 | 24.5† | 73.9 | 46.3 |
| Tülu 2 13B | 55.4 | 46.0 | 49.5 | 53.2 | 31.4 | 78.9 | 52.4 |
| Tülu 2 70B | 67.3 | **73.0** | 68.4 | 53.6 | 41.6 | 86.6 | 65.1 |
| Mistral 7B | 60.1 | 44.5 | 55.6 | 55.8 | 30.5 | 0.0 | 41.1 |
| Mistral 7B Instruct | 53.0 | 36.0 | 38.5 | 27.8 | 34.0 | 75.3 | 44.1 |
| Mixtral 8x7B Instruct | **68.4** | 65.0 | 55.9 | 24.3 | **53.5** | **94.8** | 60.3 |
| **GRITLM** | | | | | | | |
| Emb.-only 7B | 23.5 | 1.0 | 0.0 | 21.0 | 0.0 | 0.0 | 7.6 |
| Gen.-only 7B | 57.5 | 52.0 | 55.4 | 56.6 | 34.5 | 75.4 | 55.2 |
| GRITLM 7B | 57.6 | 57.5 | 54.8 | 55.4 | 32.8 | 74.8 | 55.5 |
| GRITLM 8x7B | 66.7 | 61.5 | **70.2** | 58.2 | 53.4 | 84.0 | **65.7** |

## 3.2 MAIN RESULTS

**GRIT leads to a strong embedding and generative model**   We benchmark GRITLM 7B, GRITLM 8X7B and generative- and embedding-only variants with other models in Table 1 and Table 2. We find that GRITLM 7B outperforms various prior open models on the Massive Text Embedding Benchmark (Muennighoff et al., 2023c) while still outperforming a range of generative models up to its size of 7 billion parameters. For our comparisons we focus on models that are similar to GritLM (e.g. E5 Mistral (Wang et al., 2024) uses the same base model, Instructor (Su et al., 2023) uses a similar dataset, etc.), but we note that there have been various recent embedding and generative models with stronger performance, such as Llama3 (Dubey et al., 2024), NV-Embed (Lee et al., 2024) and others (Li et al., 2024; 2023c; Meng et al., 2024; Chen et al., 2024b; Kim et al., 2024). However, GRIT models are the only ones that can handle both embedding and generation at strong performance (Figure 1). For example, using Llama 70B (Touvron et al., 2023) for embedding leads to a score of only 35.6 on MTEB as depicted in Table 1. GRITLM almost doubles that performance on MTEB, while still outperforming Llama 70B on generative tasks by more than 20% (Table 2). For GRITLM 8X7B, the embedding performance slightly decreases from GRITLM 7B, which is likely because we had to decrease its embedding batch size from 2048 for GRITLM 7B to only 256 for GRITLM 8X7B due to compute limitations (§3.1). We also train embedding-only and generative-only variants of GRITLM that only use representational or generative instruction tuning but are otherwise equivalent. Benchmarking the embedding-only variant (or models like SGPT BE 5.8B (Muennighoff, 2022)) on generative tasks in Table 2 by simply re-adding the language modeling head that was dropped during embedding finetuning leads to around random performance (25.0 is the random baseline on MMLU). Similarly, benchmarking the embedding performance of the generative-only model only leads to a score of 41.2 in Table 1. Thus, joint optimization via the GRIT approach is critical to achieve strong performance for both embedding and generation. We note, however, that with 7 billion parameters GRITLM 7B is significantly more costly to run than many other embedding models in Table 1, such as BGE Large with only 335 million parameters (Xiao et al., 2023). In addition, GRITLM 7B produces representations of 4096 dimensions, which require 4× more storage than the 1024-dimensional embeddings of BGE Large.

**GRITLM matches embedding-only and generative-only variants**   We find that unifying the two objectives via GRITLM matches both the generative-only and the embedding-only variants. This is similar to observations made for visual models (Yu et al., 2022). However, while GRITLM is trained for the same number of steps as the embedding- and generative-only models, it needs more compute per training step as it does a forward and backward pass on both embedding and generative data.

## 4 RERANKING WITH GRIT

For retrieval tasks, it is common to follow the embedding-based retrieval stage by a reranking stage (Nogueira & Cho, 2020). In the reranking stage, for each query, the top-$k$ chosen documents are reranked based on a usually more expensive but more performant method. For LLMs, prior work has shown that this can be done by passing each of the $k$ documents together with the query to the model and scoring the pair with log probabilities (Muennighoff, 2022). Note that this scales quadratically with the number of documents and queries and is thus usually too expensive for the first stage ("Cross-Encoder"). Meanwhile, using embeddings for the first stage is much cheaper as it only requires passing each query and each document once and thus scales linearly ("Bi-Encoder"). More recent work relies on instructions to use LLMs for reranking (Sun et al., 2023; Ma et al., 2023b; Pradeep et al., 2023a;b). While prior work uses separate models for the embedding and rerank-

Table 3: **Reranking (Rerank) using GRITLM as both Bi- and Cross-Encoder.**

| MTEB DS (↓) | No Rerank | Rerank top 10 |
|---|---|---|
| ArguAna | 63.24 | **64.39** |
| ClimateFEVER | 30.91 | **31.85** |
| CQADupstack | 49.42 | **50.05** |
| DBPedia | 46.60 | **47.82** |
| FiQA2018 | 59.95 | **60.39** |
| FEVER | 82.74 | **82.85** |
| HotpotQA | 79.40 | **80.46** |
| NFCorpus | 40.89 | **41.23** |
| NQ | 70.30 | **71.49** |
| MSMARCO | 41.96 | **42.47** |
| QuoraRetrieval | **89.47** | 88.67 |
| SCIDOCS | 24.41 | **24.54** |
| SciFact | 79.17 | **79.28** |
| TRECCOVID | 74.80 | **75.24** |
| Touche2020 | 27.93 | **28.41** |
| Average | 57.4 | **57.9** |

ing stages, GRITLM can be used for both stages due to its unified capabilities. In Table 3, we display the embedding performance of GRITLM 7B when additionally allowing it to rerank the top 10 documents selected via its embedding capabilities for each query. For reranking, we use the model's generative capabilities following the permutation generation approach from Sun et al. (2023) with their prompt. We find that reranking via the generative capabilities of GRITLM 7B allows it to improve on its embedding performance on most retrieval datasets. Increasing the top-$k$ documents beyond ten likely improves results, however, at the cost of more compute (Muennighoff, 2022).

## 5 RAG WITH GRIT

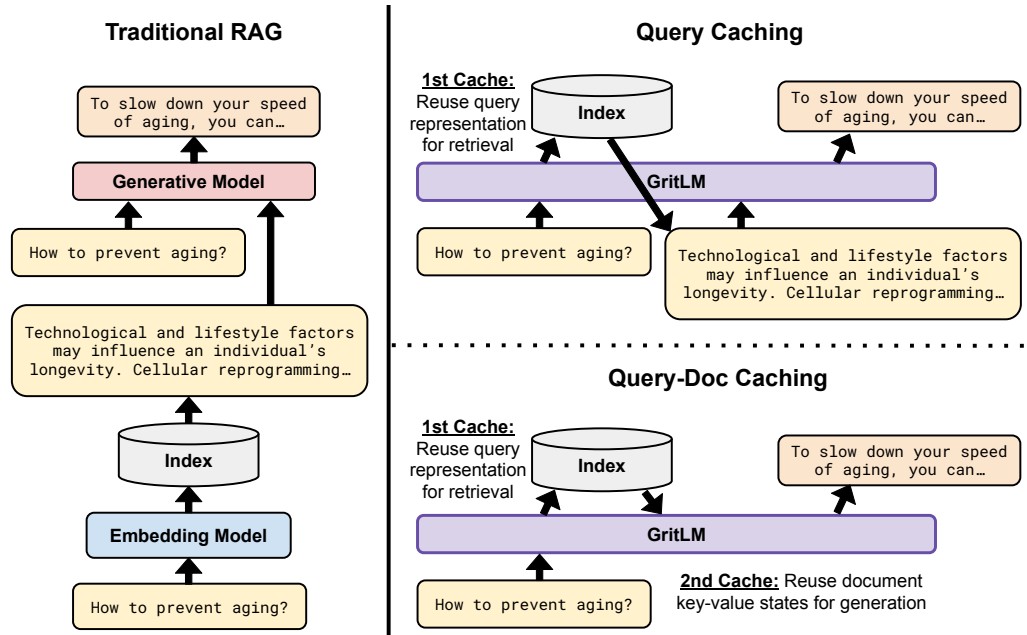

Figure 4: **RAG with GRIT.** *Left:* Traditional Retrieval-Augmented Generation (RAG) relies on a separate embedding model and generative model. *Right:* GRITLM simplifies RAG as it handles both embedding and generation. Query Caching removes the duplicate forward pass of the query by reusing its representation. Query-Doc Caching also removes the forward pass on the document during inference, as the cached index also stores the document key-value states.

**Method** By unifying embedding and generation, GRITLM simplifies Retrieval-Augmented Generation (RAG). Figure 4 displays how caching can reduce forward passes. Specifically, we introduce:
**(a) Query Caching:** In traditional RAG, the query needs to be passed both through the embedding model and later through the generative model. In Query Caching, we cache the key-value states from the embedding forward pass and reuse them for the generative pass, exploiting the property that both are the same model: GRITLM. Thus, we save compute equivalent to one forward pass of the query. Equivalently, we can also perform the generative forward pass over the query first and use its representation to retrieve the document on the fly (depicted in Figure 4). To make the generations with Query Caching completely equivalent to RAG, we place the query at the beginning of the prompt such that it only attends to itself through causal attention.
**(b) Doc Caching:** Here we cache the documents, D. When the index is created, we also save the key-value states of every document and add them to the index. Thus, the index consists of the document embeddings and key-value states. Note that the computational cost of creating the index remains the same as the key-value states have to be computed even if only embeddings are desired. At inference, we still retrieve based on embedding similarity but the index returns the key-value states instead of the text passage. These key-value states are then provided to the model to avoid having to recompute them. This effectively saves a forward pass for every in-context document at inference. However, this method increases the necessary storage. While the text passages no longer need to be

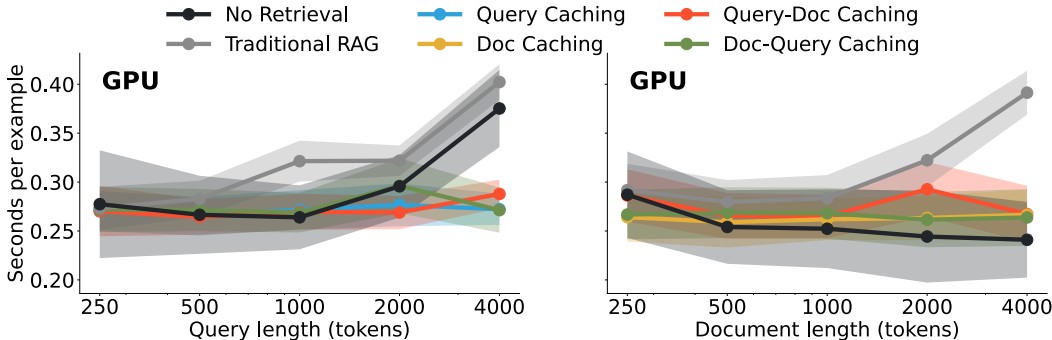

Figure 5: **Inference latency of RAG with GRITLM 7B.** When benchmarking scaling query length (left), document length is fixed at 1, whereas query length is fixed at 1 when scaling document length (right). In addition to the query/doc lengths, the formatting and prompt take up around 40 tokens. We visualize the standard deviation across 100 runs as the shaded area. For each approach, we generate 16 tokens. See Figure 6 for CPU latency.

stored, the key-value states now need to be stored and they usually require more storage depending on the model. We note that Document Caching also works for models other than GRITLM. However, for such models, one needs to pass all documents through the generation model ahead of time, thus increasing the cost of creating the index. To maintain equivalence with RAG, the document should be at the beginning of the prompt for Document Caching (opposite of Query Caching).
**(b) Query-Doc Caching / Doc-Query Caching:** We can also combine Query Caching and Doc Caching to save even more inference costs. However, combining them inevitably leads to discrepancies compared to RAG, as in traditional RAG either the query or the document is conditioned on the other one. Meanwhile, if both are cached then they are not conditioned on one another via the self-attention mechanism. We refer to Query-Doc Caching if the query is followed by the document in the prompt and to Doc-Query Caching if the document comes first.

**Setup** We benchmark the caching variants on Natural Questions (Kwiatkowski et al., 2019) using 2,681,468 documents from BEIR NQ (Thakur et al., 2021) as our index. We score models by checking if any correct answer is anywhere in the generation ("match"). Prior work often checks if the generation exactly matches the answer ("exact match") (Izacard et al., 2022). However, due to the chat data our model answers in few sentences, thus exact match fails to credit many correct answers. In the first 20 samples of the "No RAG" baseline, "exact match" leads to 4 false negatives that "match" credits correctly without any false positives. We do not use instructions for embedding here, only the format in Figure 3.

**Performance** As depicted in Table 4, RAG performs better than the "No RAG" baseline where the model is not provided any context. This validates that despite its small size compared to prior work (Lin et al., 2023), our index is still valuable. While Query and Doc Caching can theoretically lead to the exact same performance as RAG, we experience differences for two reasons: **1) Attention:** Our model is trained to embed with bidirectional attention (§2) and thus we use bidirectional attention when embedding query or document. Meanwhile, the generative model expects causal key-value states. In the Query-Doc/Doc-Query setup, there is an additional mismatch in either the documents or the queries not having attended to the other one, as both need to be embedded and cached separately. **2) Formatting:** The query is formatted in the embedding format as depicted in Figure 3, which the model has never seen during generative training. This could further lead to a performance drop. Due to 1) and 2), Query Caching leads to a performance drop compared to traditional RAG. However, the Query Caching performance of 25.46 is still better than not using RAG, thus it comes down to a speed-performance trade-off. Formatting the RAG baseline using the embedding format (Figure 3) reduces its score from 30.50 to 29.36 (not depicted), thus the additional four-point discrepancy of Query Caching and the majority of the damage is because of the attention issue. Meanwhile, Doc Caching slightly improves performance resulting in the best match score among all methods considered. This is possibly because, unlike the query, the document does not need to be as thoroughly understood, and skimming it may suffice. Thus, the slightly corrupted key-value states do not result in

Table 4: **RAG benchmarking on Natural Questions with GRITLM 7B.** For RAG, the retrieved context is simply placed in the context of the language model in contrast to our caching alternatives (Figure 4). CPU and GPU latencies are measured on an "Intel(R) Xeon(R) Platinum 8481C CPU @ 2.70GHz" and one "NVIDIA H100 80GB HBM3", respectively. Sample A has a query of 1 token and a document of 4000 tokens, and sample B is the inverse. For each approach, we generate 16 tokens. Storage consists of the index and passages, except for Doc Caching variants where it is the index and key-value states. The index is stored in float32, while key-value states are stored in bfloat16. See Appendix G for experiments on TriviaQA and MMLU.

| | Match (0-shot, ↑) | CPU Latency (s, ↓) | | GPU Latency (s, ↓) | | Storage (↓) |
| | | Sample A | Sample B | Sample A | Sample B | |
|---|---|---|---|---|---|---|
| No RAG | 21.00 | 4.3 ± 0.36 | 13.69 ± 1.0 | 0.24 ± 0.04 | 0.38 ± 0.04 | **0GB** |
| *Query then document prompt* | | | | | | |
| RAG | 30.50 | 11.64 ± 0.74 | 14.88 ± 0.87 | 0.39 ± 0.02 | 0.40 ± 0.02 | 43GB |
| Query Caching | 25.46 | 18.30 ± 0.76 | 6.87 ± 0.89 | 0.44 ± 0.03 | **0.27 ± 0.02** | 43GB |
| Query-Doc Caching | 21.63 | **5.12 ± 0.23** | 6.62 ± 0.97 | 0.27 ± 0.03 | 0.29 ± 0.01 | 30TB |
| *Document then query prompt* | | | | | | |
| RAG | 30.47 | 14.18 ± 1.01 | 15.33 ± 0.87 | 0.39 ± 0.01 | 0.4 ± 0.01 | 43GB |
| Doc Caching | **33.38** | 5.25 ± 0.34 | 23.23 ± 1.05 | 0.27 ± 0.03 | 0.45 ± 0.02 | 30TB |
| Doc-Query Caching | 18.39 | 5.23 ± 0.37 | **6.41 ± 0.96** | **0.26 ± 0.03** | **0.27 ± 0.02** | 30TB |

a performance drop. Query-Doc and Doc-Query Caching only perform near the "No RAG" baseline in our experiments, which may limit their usefulness in practice. This is likely caused by the additional attention mismatch that they introduce. This issue as well as the formatting issue could likely be solved by an additional RAG finetuning stage on top of GRITLM, which we leave to future work.

**Latency** Caching is much faster than RAG on both CPUs and GPUs, especially for long sequences (Figure 5). In Table 4, we display that for 4000 tokens, Query Caching is 54% and 33% faster on CPUs and GPUs, respectively (Sample B). For Doc Caching it is 63% and 31% (Sample A). If going beyond 4000 tokens the speed-ups will be even larger. However, for the opposite samples in Table 4 speed remains around the same. This is because while for Sample A, Doc Caching caches 4000 tokens, for Sample B it caches only 1 token, which does not provide any speed-up. Thus, Doc Caching should be used when documents are expected to be very long, while Query Caching should be used when queries are expected to be very long. In a production setting, a simple input length check could switch from one caching mode to the other. As is the case in Table 4, caching can match or even be faster than not using retrieval at all ("No RAG"). This could be due to the embedding forward pass not using the language modeling head. For Query Caching, the language modeling head is only used for the tokens that are generated, while for "RAG" and "No RAG" it is used for the entire input. The matrix multiplication with the language modeling head is computationally expensive due to its high dimensionality, which could cause the slower speed of the no retrieval baseline. Query-Doc Caching and Doc-Query Caching cache both documents and queries and thus lead to major speed-ups for both Sample A and Sample B in Table 4. Overall, speed-ups are larger on CPUs, as GPUs can process the entire sequence in parallel, thus the advantage of caching parts of it is smaller. We also note that our RAG baseline uses our 7B parameter model for both the embedding and generative model but without caching. In practice, it is often common to have an embedding model that is much smaller and cheaper than the generative model. Nonetheless, as caching with GRITLM-7B approaches the No RAG latency in Table 4, we still expect it to be faster than setups with smaller embedding models for long sequences. In addition, it would lead to significantly better performance in that case due to the state-of-the-art retrieval performance of GRITLM.

**Storage** In most RAG setups the embeddings of all documents are precomputed and stored to be later used at inference. This is referred to as the index. In traditional RAG, the documents themselves still need to be stored, as the index is only used to find the document ID, which is then used to fetch the document text and pass it to the generative model. For Doc Caching variants documents no longer need to be stored, however, the key-value states need to be stored. The key-value states take up a lot of storage, as they consist of two tensors of shape (batch size, number of heads, sequence length, dimension per head) for each batch. For our 2,681,468 documents and the 7-billion parameter

GRITLM model, this leads to 30TB of key-value states. However, unlike the index, the key-value states can be fully offloaded to disk and do not need to be kept in memory. Once the document ID has been determined via the index, the corresponding key-value state can be simply loaded from disk. For a single sample, this corresponds to loading 12.5MB of key-value states into memory.

## 6 RELATED WORK

The story of text embedding and text generation has been a story of unification.

**Embedding Models** used to focus on word representations (Pennington et al., 2014; Mikolov et al., 2013) that struggled generalizing to full sentences or passages (Conneau & Kiela, 2018). InferSent (Conneau et al., 2018), SBERT (Reimers & Gurevych, 2019), and other models (Ni et al., 2021b;a) enabled good embeddings of both words and sentences by considering context when present. However, for optimal performance, they need separate models for symmetric and asymmetric tasks (Muennighoff et al., 2023c; Neelakantan et al., 2022). Symmetric embedding tasks are ones where the query and document come from the same distribution, such as STS. Meanwhile, for asymmetric tasks, they come from different distributions and, as such, may have very different sequence lengths like in retrieval. For example, the MTEB benchmark (Muennighoff et al., 2023c) showed that SentT5 (Ni et al., 2021b) only performs well at symmetric tasks, while GTR (Ni et al., 2021a) only at asymmetric tasks, despite both using the same base model T5 (Raffel et al., 2023). Recent embedding models unify symmetric and asymmetric tasks into one model by differentiating them in the prompt (Xiao et al., 2023; Wang et al., 2022a). Further, adding detailed instructions in the prompt has allowed unifying practically any embedding task into a single model (Su et al., 2023).

**Generative Models** used to be tailored to a single task, such as translation (Sutskever et al., 2014) or question answering (Yin et al., 2016). McCann et al. (2018) cast multiple generative tasks as question answering to unify them in a single model, but performance was limited and did not generalize to arbitrary tasks. Large-scale self-supervised pretraining has enabled the use of a single large language model (LLM) for practically any generative task (Brown et al., 2020; Chowdhery et al., 2022; Rae et al., 2022; BigScience Workshop et al., 2023; Scao et al., 2022; Groeneveld et al., 2024; Allal et al., 2023; Li et al., 2023a; Soldaini et al., 2024). However, using an LLM without careful prompting often leads to poor results (Rubin et al., 2022; Min et al., 2022b). Finetuning LLMs on instructions has emerged as a method to significantly ease the usage of LLMs to apply them to any generative task with strong performance (Wei et al., 2022; Sanh et al., 2022; Min et al., 2022a; Wang et al., 2022c; Mishra et al., 2022; Iyer et al., 2023; Üstün et al., 2024; Singh et al., 2024; Zhou et al., 2023).

The two streams of embedding and generative models have each been unified into a single model that handles any task within its stream. Unifying the two streams into a single model that handles any task both for embedding and generation is the natural next step toward a general multi-task model. SGPT (Muennighoff, 2022) was an early work in that direction. SGPT only changes 0.01% of the parameters of a large language model via BitFit (Zaken et al., 2022) to adapt it to produce well-performing embeddings. Thus, one only needs to change this small amount of parameters to switch from one stream to the other. However, SGPT still required separate asymmetric and symmetric models and did not consider the full breadth of embedding tasks. GRITLM addresses these deficiencies. GRITLM does not require switching out biases, leverages instructions to handle asymmetric or symmetric use cases, and considers the full breadth of embedding and generative tasks.

## 7 CONCLUSION

We present GRIT to unify text embedding and generation, and thus all text-based language problems, into one model: GRITLM. GRITLM 7B performs strongly on the Massive Text Embedding Benchmark, while simultaneously possessing generative capabilities that exceed some larger models. Notably, its performance matches otherwise equivalent embedding-only and generative-only variants allowing us to unify them at no performance loss. We show that GRIT simplifies the field using the examples of reranking and RAG. For reranking, we are able to improve retrieval performance by around 10% by reusing GRITLM as reranker instead of having to rely on a separate model. For RAG, we unify the retriever and reader into a single model, GRITLM, speeding up inference by >60% for long texts at no performance loss via GRIT Doc Caching. We believe GRIT paves the way for a paradigm shift in language modeling, where embedding and generation seamlessly coexist in a single model. As such, we highlight the various limitations of this work and point the community to potential future research in Appendix R.

## ACKNOWLEDGEMENTS

We thank everyone at Contextual AI, the University of Hong Kong, and Microsoft for their support. We are grateful to Hamish Ivison and Yizhong Wang for help with using the Tülu 2 repository. We thank Akari Asai, Alexander M. Rush, Brian Lester, Colin Raffel, Danqi Chen, Derek Tam, John X. Morris, Haokun Liu, Hong Liu, Mengzhou Xia, Michael Matena, Muqeeth Mohammed, Omar Khattab, Shayne Longpre, Tengyu Ma, Teven Le Scao, and Tianyu Gao for discussions and feedback.

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

# APPENDIX

# Contents

## A  CONTRIBUTIONS

Niklas Muennighoff conceived of the project and led its execution. He designed the experiments and implemented, trained, and evaluated all models, and wrote most of the paper. Hongjin Su created MEDI2, implemented and ran reranking, and helped with experiments. Liang Wang ran several dataset ablations and contributed to framing. Nan Yang, Furu Wei, Tao Yu, Amanpreet Singh, and Douwe Kiela advised the project. All authors helped edit the paper.

## B  ABLATIONS

**Attention and pooling**   We train GRITLM starting from a pretrained decoder language model which has been trained with causal attention. Prior work has shown that while embeddings of causal LLMs are competitive, they are outperformed by BERT-like encoders with bidirectional attention at the same number of parameters (Muennighoff, 2022; Devlin et al., 2019). This lines up with intuition, as bidirectional attention allows the model to adjust the representation of the first tokens based on information obtained from future tokens. Meanwhile, causal attention only allows information to propagate one way. Thus, for causal attention early tokens may yield poor representations due to a lack of understanding of the entire sample. To counter this issue, we experiment with adapting the model during finetuning to learn to use bidirectional attention. In Table 5 we find that **adapting the causally pretrained LLM with bidirectional attention provides the best embedding performance**. For fully causal embeddings, we confirm findings from Muennighoff (2022) that position-weighted mean pooling ("Wmean") leads to better embedding performance than taking the embedding of the last token despite recent work finding the opposite (Zhang et al., 2023; Ma et al., 2023a). For last token pooling, we follow Zhang et al. (2023) and use a special token. We find that adapting the model to be a PrefixLM (Raffel et al., 2023), whereby the attention over the generative instruction is bidirectional but still causal for the response ("Sample") worsens performance in contrast to prior work (Wang et al., 2022b). Thus, we stick with fully causal generation. The unified variant significantly outperforms the embedding-only variants, while underperforming the best generative-only variant. However, once we switched from MEDI to the E5 dataset in later ablations the embedding-only variant matched the unified variant. Meanwhile, the worse generative performance of the unified model was due to a suboptimal loss setting that we fixed in the loss ablations. Several papers after the initial preprint release of this work have confirmed the benefit of bidirectional attention (BehnamGhader et al., 2024; Springer et al., 2024).

**Base model**   The GRITLM approach generalizes to any generative language model, thus we ablate initializing from GPT-J 6B (Wang & Komatsuzaki, 2021), Llama 2 7B or Mistral 7B (Jiang et al., 2023). Using Mistral 7B leads to the best performance for both embedding and generative tasks. For generative tasks, this is expected as the pretrained Mistral 7B performs the best among the three (Table 2). However, for embedding tasks, GPT-J outperforms Mistral 7B (Table 1). Thus, **the embedding performance of a pretrained model is not predictive of its embedding performance after finetuning**. Rather, its generative performance appears to be a more reliable indicator of its embedding performance after finetuning.

**Generative dataset**   We benchmark our filtered Tülu 2 introduced in §3.1 (Ivison et al., 2023) with UltraChat (Ding et al., 2023; Tunstall et al., 2023) and the OpenAssistant version from Oc-toPack (Muennighoff et al., 2023a; Köpf et al., 2023; Longpre et al., 2023). Using Tülu 2 leads to better performance on every generative task considered (see Appendix J for per-task results). This is likely due to Tülu 2 containing a larger diversity of tasks (Ivison et al., 2023). Another possible reason is that Tülu 2 may have been carefully tuned on the generative evaluation datasets, as we use largely the same evaluation setup as the creators of Tülu 2 (Ivison et al., 2023).

**Embedding dataset**   We benchmark MEDI (Su et al., 2023), a new version of MEDI with better negatives which we build and call MEDI2, and the E5 dataset (Wang et al., 2024). While MEDI and MEDI2 always preface instructions with "Represent" (see e.g. Figure 11), the E5 dataset places no constraint on the instruction prefix (see e.g. Figure 12). Thus, when using the E5 dataset the "`<|embed|>`" formatting is critical to tell the model that it will be subject to the representation loss, not the generative loss (Figure 3). Further, MEDI and MEDI2 always contain instructions for both queries and documents, which we refer to as **two-sided instructions**. Meanwhile, the E5 dataset

| Attention Emb | | Attention Gen | | Pooling | Emb | Gen |
|---|---|---|---|---|---|---|
| Instruction | Sample | Instruction | Sample | | | |
| *Embedding Only* | | | | | | |
| Causal | | | | Wmean | 60.0 | - |
| Causal | Bidirectional | | | Mean | 61.0 | - |
| Bidirectional | | | | Mean | 61.8 | - |
| *Generative Only* | | | | | | |
| | | Causal | | | - | **55.2** |
| | | Bidirectional | Causal | | - | 50.7 |
| *Unified* | | | | | | |
| Causal | | Causal | | Last token | 61.2 | 53.0 |
| Causal | | Causal | | Wmean | 62.8 | 52.8 |
| **Bidirectional** | | **Causal** | | **Mean** | **64.0** | 52.9 |

(a) Attention and pooling ablations. Wmean is position-weighted mean pooling (Muennighoff, 2022).

| Variant | Emb | Gen |
|---|---|---|
| **Mistral 7B** | **54.6** | **22.4** |
| Llama 2 7B | 48.2 | 20.8 |
| GPT-J 6B | 51.9 | 14.0 |

(b) Base model

| Dataset | Emb |
|---|---|
| MEDI | 64.0 |
| MEDI2 | 64.7 |
| **E5** | **66.0** |

(c) Embedding dataset

| Dataset | Gen |
|---|---|
| **Tülu 2** | **55.2** |
| OASST | 37.7 |
| UltraChat | 47.4 |

(d) Generative dataset

| Variant | Emb | Gen |
|---|---|---|
| **No head** | **62.7** | **49.2** |
| $\rightarrow$ 1024 | 62.1 | 48.0 |

(e) Embedding head

| BS Emb:Gen | Emb | Gen |
|---|---|---|
| 256:256 | 63.2 | **53.4** |
| 4096:256 | **64.2** | 53.3 |

(f) Batch size (BS)

| Precision | Emb | Gen |
|---|---|---|
| FP32 | 66.3 | 52.4 |
| **BF16**[*] | **66.5** | **55.0** |

(g) Precision

| IBN origin | Emb | Gen |
|---|---|---|
| Any dataset | **66.0** | 50.9 |
| **Same dataset** | **66.0** | **51.1** |

(h) In-batch negatives (IBN)

| Format | Gen |
|---|---|
| **Tülu 2** | **55.2** |
| Zephyr $\beta$ | 49.0 |

(i) Format

| Tokens | Emb | Gen |
|---|---|---|
| 512 | 64.1 | 52.2 |
| **2048** | **64.7** | **53.8** |

(j) Emb training max tokens

| Gen loss type | $\mathcal{L}_{\text{Rep}}/\mathcal{L}_{\text{Gen}}$ | Emb | Gen |
|---|---|---|---|
| Token | 2.4 | 66.1 | 54.4 |
| Token | 6.0 | 66.5 | 55.0 |
| **Mix (32 $\rightarrow$ 8)** | 4.1 | **66.7** | **55.4** |

| Gen loss type | AlpacaEval |
|---|---|
| Mix (4 $\rightarrow$ 64) | 67.6 |
| **Mix (32 $\rightarrow$ 8)** | **74.7** |

(k) Loss ablations. $\mathcal{L}_{\text{Rep}}/\mathcal{L}_{\text{Gen}}$ is the loss ratio of the 1st step adjusted via $\lambda_{\text{Rep}}$ and $\lambda_{\text{Gen}}$. Mix refers to mixing sample and token level loss, e.g. (32 $\rightarrow$ 8) is token level loss across 32 samples and then sample level loss across 8 sub-batches for a total batch size of 256.

Table 5: **GRIT ablations.** Emb corresponds to the MTEB average, while Gen corresponds to the average across generative tasks (Appendix I). The embedding head variant "$\rightarrow$ 1024" corresponds to down-projecting the final hidden state with a linear layer from 4096 to 1024 dimensions, only for embedding tasks. BF16[*] means that some computations are still in FP32 as explained in Appendix B. The setting chosen for GRITLM is **bold**. Once an ablation was successful, we adopted its setting, thus the bold performance slightly varies from one table to the next. For example, the base model ablation (b) is done for just 100 hundred steps with sub-optimal formatting. Full results are in Appendix J.

uses **one-sided instructions** for asymmetric datasets (Muennighoff, 2022), whereby the documents receive no instructions, only the queries. The advantage of not using document instructions is that the document corpus can be encoded once and then cached and reused across a variety of tasks. During training on E5, symmetric tasks are also in a one-sided setting, but we still evaluate them in the two-sided format. This should not be a problem as the cosine similarity function we use during training is transitive: if sentence A with instruction is similar to sentence B without instruction, and sentence B without instruction is similar to sentence C with instruction, then we can confidently say that sentence A with instruction is also similar to sentence C with instruction. As depicted in Table 5, using the E5 dataset performs best by a wide margin. An inspection of samples, suggests that this is likely due to its superior hard negatives and diversity of tasks generated by GPT-4 (Appendix T). For our final runs with the E5 dataset, we additionally add scientific data (§3.1).

**Embedding head**  The cost of caching the embeddings of a large document corpus is directly proportional to the embedding dimensionality. To minimize such costs, we experiment with adding an embedding head consisting of a linear layer with activation that down-projects the embedding (Ni et al., 2021a; Muennighoff, 2022). This layer is only used for embedding tasks. Down-projecting the embeddings four-fold (from 4096 to 1024) leads to an embedding performance decrease of around 1%. This may be acceptable for certain use cases where the saved storage is more important. However, for our final model, we do not use such a head to keep it simple and achieve maximum performance. Search techniques (Arya et al., 1998; Johnson et al., 2017; Douze et al., 2024) or dimensionality reduction techniques such as Principal Component Analysis still allow for reducing the embedding dimension of our final model post-training while maintaining most of the performance. Similar to the storage cost-performance trade-off we explore here, we hypothesize that there is a speed/cost-performance trade-off with taking the embedding from different layers of our model. For example, we could train using the embedding after half the layers of the model, thus speeding up the embedding model by 50% while likely only incurring a small drop in embedding performance

**Batch size**  Due to the utilization of in-batch negatives for contrastive training (§2), a larger batch size provides a more accurate gradient. Thus, scaling up the batch size is a key ingredient in most well-performing embedding models (Xiao et al., 2023; Wang et al., 2022a). We experiment with scaling up the embedding batch size to 4096 while keeping it at 256 for generative data. This leads to a 1.0 gain on the embedding average while generative performance remains stable. Especially the 15 retrieval datasets that are part of the embedding average benefit from the increase in batch size (see Table 18). For our final model, we use a batch size of 2048 for embedding and 256 for generative data.

**Precision**  The parameters of the Mistral 7B model are in bfloat16 (BF16) precision as it was pretrained in this format. We experiment with finetuning it with float32 (FP32) precision versus keeping the BF16 format and training with mixed precision. FP32 training is more costly, however, the additional precision may result in a better model. Our intuition is that more precision is important for embedding but not as much for generation. This is because while for generative tasks evaluated greedily, the model output is a discretionary argmax over the predictions of the language modeling head, for embedding tasks it is a continuous representation. Thus, small differences due to a lack of precision may not change the model's generation but will affect its representation. Hence, for embedding tasks, we always cast the hidden states to FP32 during the pooling operation and keep them this way for the similarity computation. Not keeping them in FP32 after pooling worsens performance slightly, but may be necessary for cheap storage (see Appendix P). In addition, some operations such as layer normalization (Ba et al., 2016) are also performed in FP32 even for BF16 training due to PyTorch autocast (Zhao et al., 2023). In Table 5, we find that there is no benefit from doing even more computations in FP32 besides the ones listed above. Thus, we train and evaluate all our other models in BF16 mixed precision to speed up training and inference.

**In-batch negatives**  We always use in-batch negatives for embedding training (§2), however, we ablate whether or not they come from the same dataset. We hypothesize that making them all come from the same dataset leads to better negatives as the model needs to distinguish them based on more nuanced differences. In practice, we find that the average embedding performance remains around the same. However, we notice a 1.3 jump on the 15-dataset Retrieval average (Table 20). Thus, we stick with the variant where in-batch negatives stem from the same dataset.

**Format** Our chosen format is depicted in Figure 3, which is equivalent to Tülu 2 (Ivison et al., 2023) for generative tasks. We also benchmark the Zephyr $\beta$ format (Tunstall et al., 2023), which has an additional end-of-sequence token ("``") after each user utterance. We find that it performs worse on generative tasks. The additional end-of-sequence after the user utterance increases the likelihood of the model generating another end-of-sequence token earlier than necessary. This significantly harms HumanEvalSynthesize performance and slightly reduces AlpacaEval, where long generations can be critical (see Appendix J for task-specific performance).

**Max tokens** Our base model, Mistral 7B, can handle sequences of arbitrary length due to its sliding window attention (Jiang et al., 2023). As finetuning with longer sequences is more expensive we ablate its benefits. We compare training with a maximum token limit of 512 versus 2048 for embedding documents. For embedding queries, we always use 256, and for generative data, we always use 2048. We find that increasing the embedding document sequence length during training slightly boosts performance on both embedding and generation even though we still evaluate embedding tasks with 512. This boost likely comes from our training data containing many documents beyond 512 tokens, which need to be truncated if the maximum sequence length is 512. Such truncation may remove the critical parts that make two texts a positive or a negative contrastive pair and thus hinder learning. As our embedding evaluation (MTEB) contains few documents longer than 512 tokens there is little truncation happening at evaluation (Muennighoff et al., 2023c; Günther et al., 2024; 2023). Note that just like their base models, our final models GRITLM 7B and GRITLM 8X7B can produce embeddings for sequences of arbitrary length. However, due to a lack of benchmarks, we do not know how well the embeddings of our models perform for input sequences longer than 512 tokens.

**Loss ablations** As detailed in §2, we experiment with both token and sample level generative loss. Further, we ablate the representation and generative loss weights, $\lambda_{\text{Rep}}$ and $\lambda_{\text{Gen}}$. For the unified visual model CoCa, the authors find that giving a weight of 2 to generation and 1 to embedding boosts performance on both streams (Yu et al., 2022). However, rather than the weights, we argue that the loss ratio, $\mathcal{L}_{\text{Rep}}/\mathcal{L}_{\text{Gen}}$, is of more interest as it reveals which objective has a larger impact on the optimization of the model. We maintain a ratio of $\mathcal{L}_{\text{Rep}}/\mathcal{L}_{\text{Gen}}$ ¿ 1 i.e. giving more weight to the representation loss. This is because the model has already been pretrained with the generative loss, thus we expect less additional generative training to be necessary. Meanwhile, the contrastive loss for embedding data is new to the model, thus we expect more learning to be needed on the embedding side. Further, the embedding loss drops off extremely quickly as can be seen in the loss graphs in Appendix H. Thus, even though the representation loss has a higher weight at the start, throughout training they have very similar weights with both hovering around a loss of 1.0. We find that mixing sample and token level generative loss leads to the best performance by a small margin. As expected in §2, token level loss to some degree is critical for good performance on AlpacaEval. For "Mix (4 -> 64)" token level loss is applied across only 4 samples and then sample level loss across 64 sub-batches, which leads to a 7-point drop in AlpacaEval performance. This drop is accompanied by a decrease in median AlpacaEval generation length from 941 to 865. Thus, token level loss across many samples is critical to maintaining long generations, which directly impacts the AlpacaEval score.

## C DISCUSSION

**Further unification** To the best of our knowledge, GRITLM is the first model to unify text embedding and generation, and thus all text-based language problems, into a single model at strong performance. However, many adjacent directions remain to be improved or unified. **(a) Multilinguality:** Our model is also capable of embedding and generation in non-English languages as seen in its TyDi QA performance (Table 2). However, major performance gains on non-English tasks are likely possible through both data (Muennighoff et al., 2023d; Yong et al., 2023) and architecture changes (Chen et al., 2024a; Feng et al., 2022; Duquenne et al., 2023) targeting multilinguality. **(b) Multimodality:** Many embedding and generative problems are not purely text-based, such as joint embedding of images and text (Radford et al., 2021), generative image captioning (Hossain et al., 2018), image-text pair classification (Muennighoff, 2020; Kiela et al., 2021) or speech versions of every text problem (Kamath et al., 2019). It remains to be explored whether they can be as easily unified as text embedding and generation in this work.

**Why does GRIT work?** GRIT unifies embedding and generative tasks into a single model at no performance loss on either one, which may seem surprising. When the embedding dataset is MEDI2, we show that embedding performance even improves once the generative objective is added compared to an otherwise equivalent embedding-only model (Appendix B). We think that our results confirm that generative language modeling and text embeddings are two sides of the same coin. Both tasks require a model to have a deep understanding of natural language and only differ in the way that understanding is expressed. Possibly, our unified model contains a small number of parameters that act as a switch to make the final representations either useful for mean pooling and subsequent embedding tasks or primed for the language modeling head and subsequent generative tasks. We are excited about future work exploring what is happening inside of GRITLM. To support such research, we release all our work freely.

**Optimizing RAG with GRITLM** RAG and the caching variants we have presented in this work operate on a frozen language model. Meanwhile, there has been extensive work on optimizing a generative model specifically for interaction with a retrieval system (Gao et al., 2024; Zhu et al., 2024; Asai et al., 2023a). These works commonly optimize only the retriever (Shi et al., 2023) or only the reader (Borgeaud et al., 2022; Yasunaga et al., 2023; Asai et al., 2023b; Luo et al., 2023). However, recent work has shown that jointly optimizing both models leads to the best performance (Lin et al., 2023). With its state-of-the-art retrieval and generative performance, GRITLM can act as both the retriever and reader in a single model. Thus, optimizing either one also changes the parameters of the other. This has the potential to significantly simplify the joint optimization of the retriever and reader. For example, it may suffice to only use the next-token objective (Equation 2) to penalize the retriever for providing irrelevant context and at the same time the reader for poor use of the given context. This is in contrast to separate models and objective functions used in Lin et al. (2023).

## D ALIGNING GRITLM

Table 6: **Aligning GRITLM with KTO after GRIT.** The upper table depicts embedding performance while the lower depicts generative performance.

| Task (→) | CLF | Clust. | PairCLF | Rerank | Retrieval | STS | Summ. | Avg. |
|---|---|---|---|---|---|---|---|---|
| Metric (→) | Acc. | V-Meas. | AP | MAP | nDCG | Spear. | Spear. | |
| Dataset # (→) | 12 | 11 | 3 | 4 | 15 | 10 | 1 | 56 |
| GRITLM 7B | 79.5 | 50.6 | 87.2 | 60.5 | 57.4 | 83.4 | 30.4 | 66.8 |
| GRITLM 7B KTO | 79.6 | 50.1 | 87.1 | 60.5 | 57.1 | 83.5 | 30.5 | 66.7 |
| GRITLM 8X7B | 78.5 | 50.1 | 85.0 | 59.8 | 55.1 | 83.3 | 29.8 | 65.7 |
| GRITLM 8X7B KTO | 78.7 | 50.0 | 84.4 | 59.4 | 54.1 | 82.5 | 30.8 | 65.2 |

| Dataset (→) | MMLU | GSM8K | BBH | TyDi QA | HumanEval | Alpaca | Avg. |
|---|---|---|---|---|---|---|---|
| Setup (→) | 0 FS | 8 FS, CoT | 3 FS, CoT | 1 FS, GP | 0 FS | 0 FS, 1.0 | |
| Metric (→) | EM | EM | EM | F1 | pass@1 | % Win | |
| GRITLM 7B | 57.6 | 57.5 | 54.8 | 55.4 | 32.8 | 74.8 | 55.5 |
| GRITLM 7B KTO | 57.6 | 57.5 | 55.4 | 55.8 | 31.5 | 86.7 | 57.4 |
| GRITLM 8X7B | 66.7 | 61.5 | 70.2 | 58.2 | 53.4 | 84.0 | 65.7 |
| GRITLM 8X7B KTO | 66.8 | 79.5 | 67.1 | 31.4 | 56.8 | 95.3 | 66.2 |

It is common to follow the instruction finetuning stage of generative language models by an alignment tuning stage using methods like PPO (Schulman et al., 2017), DPO (Rafailov et al., 2023), or KTO (Ethayarajh et al., 2024) ("HALOs" (Ethayarajh et al., 2024)). We experiment with further finetuning GRITLM using KTO and benchmark the resulting models in Table 6. During this KTO stage, no further embedding training is performed, thus it leads to a slight performance drop on the MTEB average (66.8 to 66.7 and 65.7 to 65.2). However, the average generative performance of the KTO-tuned models is stronger. Notably, AlpacaEval jumps by ¿10 points for both models. On the more recent Alpaca 2.0 (Dubois et al., 2024), GritLM-8x7B-KTO has a length-controlled win rate of 18.5 with an average length of 1662 (not depicted). Thus, the KTO-finetuned models may be more useful for use cases where the generative performance is more important. Future work may consider continuing the embedding training during the alignment tuning stage. It may also be possible to

develop an alignment tuning method specifically for embedding performance and combine it with generative alignment via KTO.

## E  FEW-SHOT EMBEDDING DOES NOT WORK

For generative models it has been well-established that providing in-context examples ("few-shots", FS) improves performance (Brown et al., 2020). However, to the best of our knowledge, there has been no work on in-context learning with embedding models. In Table 7, we benchmark the default 0-shot format with providing a single few-shot example following the task instruction. We take the few-shot example from the respective evaluation dataset (see §U.2 for the prompts). We find that providing few-shot examples overall worsens performance. While there are small gains among PairClassification tasks (SprintDup. and TwitterURL), these are marginal and inconsistent. For the model trained on MEDI2, we even include few-shot embedding samples in the training data for around 5% of training samples. However, the model seems not to have learned to make good use of the few-shot examples.

Table 7: **Few-shot embedding.** The 12 MTEB datasets ("DS") are grouped by the 7 main MTEB tasks in the same order as in Table 1.

| Train DS (→) | E5S | | MEDI2 | |
|---|---|---|---|---|
| MTEB DS (↓) | 0 FS | 1 FS | 0 FS | 1 FS |
| Banking77 | **88.5** | 88.3 | **88.1** | 87.9 |
| Emotion | **52.8** | 51.0 | **52.5** | 51.9 |
| IMDB | **95.0** | 93.9 | **94.3** | 92.2 |
| BiorxivS2S | **39.8** | 39.4 | **37.6** | 37.4 |
| SprintDup. | 93.0 | **94.9** | 95.2 | **95.7** |
| TwitterSem | **81.1** | 77.9 | **76.8** | 73.9 |
| TwitterURL | **87.4** | 87.1 | 85.9 | **86.1** |
| ArguAna | **63.2** | 51.7 | **53.5** | 53.2 |
| SCIDOCS | **24.4** | 19.7 | **25.5** | 25.5 |
| AskUbuntu | **67.3** | 64.7 | **66.6** | 66.0 |
| STS12 | 77.3 | **78.0** | **76.6** | 73.5 |
| SummEval | **30.4** | 29.5 | 29.1 | **31.5** |

## F  RAG CACHING CPU LATENCY

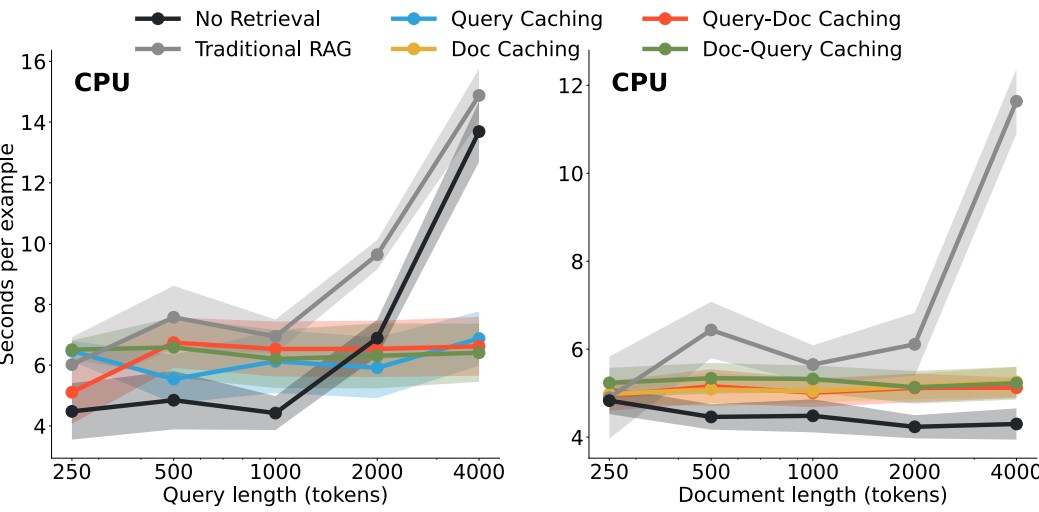

Figure 6: **Inference latency of RAG with GRITLM 7B on CPUs.** When benchmarking scaling query length (left), document length is fixed at 1, whereas query length is fixed at 1 when scaling document length (right). In addition to the query/doc lengths, the formatting and prompt take up around 40 tokens. We visualize the standard deviation across 100 runs as the shaded area. For each approach, we generate 16 tokens. See Figure 5 for GPU latency.

## G  ADDITIONAL RAG RESULTS

In §5, we find that doc caching is the most promising caching variant out of the ones we propose. This is because **(a)** documents are usually significantly longer than queries, thus caching documents has the highest potential to reduce latency, **(b)** it maintains performance of regular RAG (Table 4, and **(c)** it even works for non-GRIT models though it requires more time to construct the cache for non-GRIT models (§5). Thus we further experiment with doc caching in Table 8 to verify its performance on other datasets. Similar to Natural Questions in Table 4, we observe that doc caching maintains performance of regular RAG (even slightly improves) for TriviaQA and MMLU despite the attention mismatch. Note that the attention mismatch problem can

Table 8: **Additional doc caching results.** We use the same setup as in Table 4 to benchmark doc caching on two additional datasets: TriviaQA (Joshi et al., 2017) and MMLU (Hendrycks et al., 2022).

| Dataset (→) | TriviaQA | MMLU |
|---|---|---|
| Metric (→) | Match (0-shot, ↑) | |
| RAG | 52.12 | 51.10 |
| Doc Caching | **57.93** | **53.46** |

always be resolved by simply not using bidirectional attention for the embedding part and thereby guarantee the same performance as not using RAG, however, not using bidirectional attention comes at a slight reduction in embedding performance according to our ablation experiments (Appendix B).

We also benchmark the BGE series of embedding models (Xiao et al., 2023) in Table 9 for RAG. We find performance to be significantly worse than with GRITLM in Table 4. Based on a manual inspection of samples, it appears that the embedding models commonly retrieve irrelevant passages that confuse the generative model. There may be other smaller embedding models or other generative models that may perform better, but overall we expect the RAG performance to be a function of the embedding and generative performance of the individual components (e.g. if an embedding model performs better than GRITLM, we would expect it to lead to better RAG performance; BGE generally does not perform better on embedding as shown in Table 1).

Table 9: **Additional RAG results with BGE.** We use the same setup as in Table 4 to benchmark BGE embedding models with the "Query then document" prompt. The generative model is still GRITLM 7B.

| Dataset (→) | NQ |
|---|---|
| Metric (→) | Match (0-shot, ↑) |
| BGE Large 0.34B | 10.39 |
| BGE Base 0.11B | 10.31 |
| BGE Small 0.03B | 10.17 |

## H  LOSS CURVES

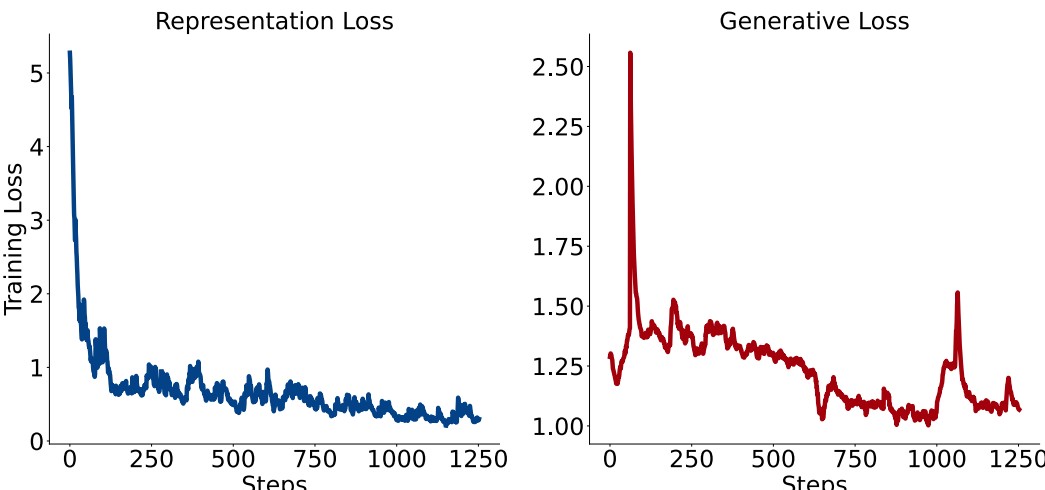

Figure 7: **GRITLM 7B training loss smoothed with exponential moving average smoothing and a weight of 0.9.**

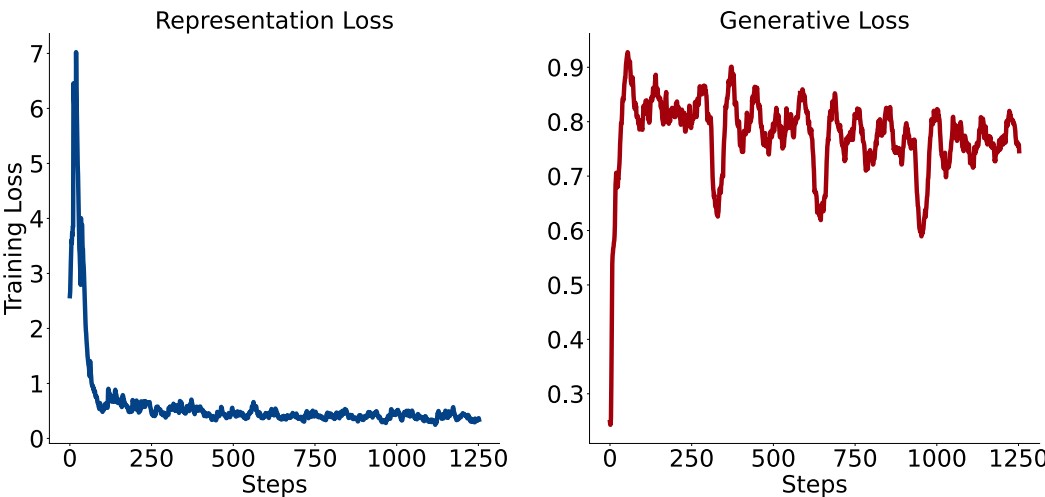

Figure 8: **GRITLM 8X7B training loss smoothed with exponential moving average smoothing and a weight of 0.9.**

# I  EVALUATION

For evaluating GRITLM, we select the most commonly used embedding and generative benchmarks:

**Embedding**   To evaluate embedding performance we use the 7 main tasks from MTEB (Muennighoff et al., 2023c).
**(1) Classification (CLF):** A logistic regression classifier is trained on embeddings from texts with different labels. The classifier is scored with **F1**.
**(2) Clustering (Clust.):** K-means clustering is performed on embeddings from different sources. The agreement of the clusters with respect to the source labels is scored with **V-measure**.
**(3) Pair Classification (PairCLF):** The cosine similarity of two embeddings with a binary label is computed. The optimal similarity threshold across all samples is found and scored with **AP** (average precision).
**(4) Reranking (Rerank)** A query embedding and reference embeddings are compared with cosine similarity. The similarities are scored versus the ground truth ranking of the references via **MAP** (mean AP).
**(5) Retrieval:** A query embedding and embeddings of references are compared with cosine similarity. The position of the correct reference(s) in the top ten with the highest cosine similarity is scored with **nDCG@10** (normalized discounted cumulative gain).
**(6) STS:** The cosine similarity of two embeddings is compared with a ground truth continuous score of their similarity and scored with **Spearman** correlation.
**(7) Summarization (Summ.)** Human-written and machine-written summaries of the same text are embedded. The cosine similarity of the embeddings is compared to human ratings of the machine summaries and scored with **Spearman** correlation.
Among the tasks, Reranking, Retrieval, and Summarization are asymmetric i.e. there are two different kinds of embeddings: queries and documents. Others are symmetric i.e. there is only one kind. We use instructions for every dataset specified in §U.1. Notably, for some models, we use different instructions for query and document embeddings when dealing with asymmetric tasks. The datasets within each task cover diverse domains ranging from scientific papers to casual conversations.

**Generation**   For evaluating the generative performance of GRITLM, we largely follow the evaluation setup of Tülu (Wang et al., 2023; Ivison et al., 2023) using open-source frameworks (Gao et al., 2021a; Ben Allal et al., 2022).
**(1) Multiple-Choice Question Answering via MMLU (Hendrycks et al., 2022):** Models are tasked to answer knowledge-intensive questions from different fields, such as humanities, social sciences, and hard sciences. No few-shots are provided and answers are evaluated with **exact match**.

**(2) Problem solving via GSM (Cobbe et al., 2021):** Models are tasked to solve a math problem requiring multi-step reasoning. 8 few-shot (FS) examples with chain-of-thought reasoning (CoT) (Wei et al., 2023) are provided and **exact match** is measured.

**(3) Multilingual Closed-book Question Answering via TyDi QA (Clark et al., 2020):** Models are tasked to answer a question in one of six languages. We evaluate in the Gold Passage and no-context setting following Anil et al. (2023).

**(4) Code Generation via HumanEvalSynthesize (Muennighoff et al., 2023a; Chen et al., 2021):** We use the HumanEvalSynthesize Python dataset (Muennighoff et al., 2023a), which is adapted from HumanEval (Chen et al., 2021) for easy evaluation of instruction-following models. Using the instruction format is different from Ivison et al. (2023) who use HumanEval without an instruction format which is not how the model is used in practice. Following Muennighoff et al. (2023a), we score pass@1 using 20 samples and a temperature of 0.2.

**(5) Boolean Expressions, Causal Judgement, etc. via BBH (Srivastava et al., 2023; Suzgun et al., 2022)** We evaluate a variety of reasoning tasks using BIG-Bench Hard (BBH) (Srivastava et al., 2023; Suzgun et al., 2022). Similar to GSM8K, 3 FS CoT examples are provided and **exact match** is measured.

**(6) Open-ended writing, Summarization, Role-playing, etc. via AlpacaEval (Alpaca) (Li et al., 2023b; Dubois et al., 2023)** We evaluate a variety of open-ended generation tasks via the original 1.0 version of AlpacaEval (Li et al., 2023b; Dubois et al., 2023). GPT-4 (OpenAI et al., 2023) is used to determine the win rate of generations compared to provided GPT-3 (Brown et al., 2020) answers. We differ from Ivison et al. (2023) in that we reduce the maximum token length to 6144 from 8192. We do not use MT-Bench due to its limitations pointed out in Appendix Q. To ensure reproducibility, we use greedy evaluation throughout.

## J  ABLATIONS DETAILED RESULTS

We display a breakdown of the results from Table 5 in Table 10 to Table 21. For MTEB per-dataset results, we refer to Appendix K, the MTEB leaderboard (`https://huggingface.co/spaces/mteb/leaderboard`) and our released result files (`https://huggingface.co/datasets/GritLM/results`).

Table 10: **Unified models attention and pooling ablations.** The sequence of Cs and Bs refers to the attention mechanism for (from left to right): Emb instruction, Emb sample, Gen instruction, Gen sample, where C=Causal, B=Bidirectional, Emb=Embedding and Gen=Generative. WM, LT and M refer to position-weighted mean, last token and mean pooling, respectively.

| Task (→) | CLF | Clust. | PairCLF | Rerank | Retrieval | STS | Summ. | Avg. |
|---|---|---|---|---|---|---|---|---|
| Metric (→) | Acc. | V-Meas. | AP | MAP | nDCG | Spear. | Spear. | |
| Dataset # (→) | 12 | 11 | 3 | 4 | 15 | 10 | 1 | 56 |
| CCCC WM | 77.9 | 47.9 | 81.5 | 59.0 | 49.4 | 80.3 | 29.4 | 62.8 |
| CCCC LT | 78.8 | 46.9 | 84.5 | 59.6 | 43.9 | 78.7 | 29.3 | 61.2 |
| BBCC M | 79.0 | 48.6 | 86.3 | 59.5 | 49.9 | 81.7 | 30.1 | 63.8 |

| Dataset (→) | MMLU | GSM8K | BBH | TyDi QA | HumanEval | Alpaca | Avg. |
|---|---|---|---|---|---|---|---|
| Setup (→) | 0 FS | 8 FS, CoT | 3 FS, CoT | 1 FS, GP | 0 FS | 0 FS, 1.0 | |
| Metric (→) | EM | EM | EM | F1 | pass@1 | % Win | |
| CCCC WM | 57.5 | 45.0 | 53.1 | 56.0 | 32.3 | 72.9 | 52.8 |
| CCCC LT | 57.2 | 45.5 | 54.7 | 54.0 | 31.1 | 75.7 | 53.0 |
| BBCC M | 57.0 | 46.5 | 54.5 | 55.0 | 30.4 | 73.8 | 52.9 |

Table 11: **Embedding-only models attention and pooling ablations.** The sequence of Cs and Bs refers to the attention mechanism for (from left to right): Emb instruction, Emb sample, where C=Causal, B=Bidirectional and Emb=Embedding. WM and M refer to position-weighted mean and mean pooling, respectively.

| Task (→) | CLF | Clust. | PairCLF | Rerank | Retrieval | STS | Summ. | Avg. |
|---|---|---|---|---|---|---|---|---|
| Metric (→) | Acc. | V-Meas. | AP | MAP | nDCG | Spear. | Spear. | |
| Dataset # (→) | 12 | 11 | 3 | 4 | 15 | 10 | 1 | 56 |
| CC WM | 77.1 | 44.0 | 83.3 | 57.0 | 43.2 | 79.6 | 29.4 | 60.0 |
| CB M | 76.4 | 45.5 | 83.1 | 56.8 | 45.7 | 80.6 | 30.4 | 61.0 |
| BB M | 77.3 | 46.0 | 83.8 | 58.2 | 46.8 | 81.0 | 32.3 | 61.8 |

Table 12: **Generative-only models attention ablations.** The sequence of Cs and Bs refers to the attention mechanism for (from left to right): Gen instruction, Gen sample, where C=Causal and B=Bidirectional. IL=interleaved, whereby the bidirectional attention is interleaved with causal attention in multi-turn samples (bidirectional for instructions, causal for answers). This allows for faster generation in multi-turn settings as the kv-cache of the answer can be reused.

| Dataset (→) | MMLU | GSM8K | BBH | TyDi QA | HumanEval | Alpaca | Avg. |
|---|---|---|---|---|---|---|---|
| Setup (→) | 0 FS | 8 FS, CoT | 3 FS, CoT | 1 FS, GP | 0 FS | 0 FS, 1.0 | |
| Metric (→) | EM | EM | EM | F1 | pass@1 | % Win | |
| CC | 57.5 | 52.0 | 55.4 | 56.6 | 34.5 | 75.4 | 55.2 |
| BC | 57.2 | 50.0 | 49.3 | 52.0 | 30.6 | 64.8 | 50.7 |
| BC IL | 52.6 | 41.0 | 46.9 | 45.4 | - | - | - |

Table 13: **Base model ablations.** Models are only trained for 100 steps and with other sub-optimal settings, such as the Zephyr format, that were rectified through later ablations.

| Task (→) | CLF | Clust. | PairCLF | Rerank | Retrieval | STS | Summ. | Avg. |
|---|---|---|---|---|---|---|---|---|
| Metric (→) | Acc. | V-Meas. | AP | MAP | nDCG | Spear. | Spear. | |
| Dataset # (→) | 12 | 11 | 3 | 4 | 15 | 10 | 1 | 56 |
| Mistral 7B | 70.6 | 43.7 | 74.0 | 54.8 | 35.3 | 72.9 | 31.2 | 54.6 |
| Llama 2 7B | 68.1 | 38.0 | 64.1 | 50.2 | 24.2 | 67.7 | 30.5 | 48.2 |
| GPT-J 6B | 70.7 | 41.4 | 69.6 | 53.9 | 29.7 | 70.4 | 29.8 | 51.9 |

| Dataset (→) | MMLU | GSM8K | BBH | TyDi QA | HumanEval | Avg. |
|---|---|---|---|---|---|---|
| Setup (→) | 0 FS | 8 FS, CoT | 3 FS, CoT | 1 FS, GP | 0 FS | |
| Metric (→) | EM | EM | EM | F1 | pass@1 | |
| Mistral 7B | 35.0 | 11.0 | 31.6 | 20.5 | 13.8 | 22.4 |
| Llama 2 7B | 35.8 | 7.0 | 27.2 | 21.0 | 12.9 | 20.8 |
| GPT-J 6B | 27.5 | 3.5 | 22.2 | 8.7 | 8.0 | 14.0 |

Table 14: **Embedding-only models embedding dataset ablations.** NNI = No Natural Instructions, corresponding to not including natural instructions in the data. II = evaluating with the Instructor-XL instructions (Su et al., 2023). Other models use our new structure with domain, intent, and unit depicted in Figure 3. Thus, MEDI2 NNI II and MEDI2 NNI are the same model and only differ in the evaluation instruction set.

| Task (→) | CLF | Clust. | PairCLF | Rerank | Retrieval | STS | Summ. | Avg. |
|---|---|---|---|---|---|---|---|---|
| Metric (→) | Acc. | V-Meas. | AP | MAP | nDCG | Spear. | Spear. | |
| Dataset # (→) | 12 | 11 | 3 | 4 | 15 | 10 | 1 | 56 |
| MEDI II | 77.1 | 44.0 | 83.3 | 57.0 | 43.2 | 79.6 | 29.4 | 60.0 |
| MEDI2 NNI II | 74.0 | 43.5 | 80.5 | 56.6 | 46.1 | 78.4 | 29.5 | 59.6 |
| MEDI2 NNI | 74.2 | 44.5 | 80.7 | 57.3 | 49.5 | 79.6 | 30.8 | 61.1 |
| MEDI2 | 75.1 | 43.8 | 80.6 | 57.5 | 50.2 | 81.7 | 31.9 | 61.7 |
| MEDI2 + Weights | 74.4 | 42.7 | 78.4 | 57.7 | 50.2 | 81.4 | 30.5 | 61.2 |

Table 15: **Unified models embedding dataset ablations.** The sequence of Cs and Bs refers to the attention mechanism for (from left to right): Emb instruction, Emb sample, where C=Causal, B=Bidirectional, and Emb=Embedding. WM and M refer to position-weighted mean and mean pooling, respectively. MEDI2BGE corresponds to our MEDI2 dataset with negatives coming from the BGE training dataset MTP (Xiao et al., 2023).

| Task (→) | CLF | Clust. | PairCLF | Rerank | Retrieval | STS | Summ. | Avg. |
|---|---|---|---|---|---|---|---|---|
| Metric (→) | Acc. | V-Meas. | AP | MAP | nDCG | Spear. | Spear. | |
| Dataset # (→) | 12 | 11 | 3 | 4 | 15 | 10 | 1 | 56 |
| CCCC WM MEDI | 77.9 | 47.9 | 81.5 | 59.0 | 49.4 | 80.3 | 29.4 | 62.8 |
| CCCC WM MEDI2 | 76.5 | 47.0 | 82.5 | 59.4 | 51.4 | 81.9 | 30.2 | 63.2 |
| BBCC M MEDI | 79.1 | 48.8 | 86.4 | 59.6 | 50.3 | 81.3 | 31.0 | 64.0 |
| BBCC M MEDI2 | 77.0 | 48.7 | 86.0 | 61.0 | 53.6 | 83.0 | 29.1 | 64.7 |
| BBCC M MEDI2BGE | 77.0 | 48.9 | 86.9 | 61.3 | 53.1 | 82.8 | 29.4 | 64.7 |
| BBCC M E5 | 79.7 | 49.5 | 86.2 | 59.6 | 55.3 | 83.6 | 29.9 | 66.0 |

| Dataset (→) | MMLU | GSM8K | BBH | TyDi QA | HumanEval | Alpaca | Avg. |
|---|---|---|---|---|---|---|---|
| Setup (→) | 0 FS | 8 FS, CoT | 3 FS, CoT | 1 FS, GP | 0 FS | 0 FS, 1.0 | |
| Metric (→) | EM | EM | EM | F1 | pass@1 | % Win | |
| CCCC WM MEDI | 57.5 | 45.0 | 53.1 | 56.0 | 32.3 | 72.9 | 52.8 |
| CCCC WM MEDI2 | 57.1 | 49.0 | 53.3 | 55.3 | 32.3 | 73.6 | 53.4 |
| BBCC M MEDI | 57.0 | 46.5 | 54.5 | 55.0 | 30.4 | 73.8 | 52.9 |
| BBCC M MEDI2 | 57.0 | 50.5 | 53.8 | 54.7 | 32.3 | 74.7 | 53.8 |
| BBCC M MEDI2BGE | 57.4 | 48.0 | 54.7 | 55.1 | 32.0 | 74.7 | 53.7 |
| BBCC M E5 | 57.3 | 47.5 | 54.2 | 54.6 | 33.6 | 75.4 | 53.8 |

Table 16: **Generative dataset ablations.** EP = number of epochs.

| Dataset (→) | MMLU | GSM8K | BBH | TyDi QA | HumanEval | Alpaca | Avg. |
|---|---|---|---|---|---|---|---|
| Setup (→) | 0 FS | 8 FS, CoT | 3 FS, CoT | 1 FS, GP | 0 FS | 0 FS, 1.0 | |
| Metric (→) | EM | EM | EM | F1 | pass@1 | % Win | |
| Tülu 2 1 EP | 57.5 | 52.0 | 55.4 | 56.6 | 34.5 | 75.4 | 55.2 |
| Tülu 2 2 EP | 58.2 | 53.0 | 51.9 | 54.1 | 37.4 | 80.5 | 55.9 |
| OASST 1 EP | 53.8 | 24.0 | 41.1 | 28.2 | 27.4 | 51.7 | 37.7 |
| OASST 2 EP | 52.4 | 17.5 | 45.7 | 29.2 | 19.8 | 61.3 | 37.7 |
| UltraChat | 56.1 | 43.0 | 53.8 | 35.0 | 25.9 | 70.3 | 47.4 |

Table 17: **Embedding Head.** "→ 1024" refers to down-projecting the final hidden state with a linear layer from 4096 to 1024 dimensions only for embedding tasks.

| Task (→) | CLF | Clust. | PairCLF | Rerank | Retrieval | STS | Summ. | Avg. |
|---|---|---|---|---|---|---|---|---|
| Metric (→) | Acc. | V-Meas. | AP | MAP | nDCG | Spear. | Spear. | |
| Dataset # (→) | 12 | 11 | 3 | 4 | 15 | 10 | 1 | 56 |
| No head | 77.7 | 47.9 | 81.3 | 58.6 | 49.2 | 80.4 | 29.5 | 62.7 |
| → 1024 | 76.9 | 47.6 | 82.1 | 58.6 | 48.0 | 80.1 | 29.8 | 62.1 |

| Dataset (→) | MMLU | GSM8K | BBH | TyDi QA | HumanEval | Alpaca | Avg. |
|---|---|---|---|---|---|---|---|
| Setup (→) | 0 FS | 8 FS, CoT | 3 FS, CoT | 1 FS, GP | 0 FS | 0 FS, 1.0 | |
| Metric (→) | EM | EM | EM | F1 | pass@1 | % Win | |
| No head | 54.2 | 42.5 | 50.6 | 53.9 | 28.4 | 65.5 | 49.2 |
| → 1024 | 53.6 | 37.0 | 48.8 | 54.4 | 26.6 | 67.3 | 48.0 |

Table 18: **Embedding batch size ablations.** 256 and 4096 indicate the respective embedding batch size. The generative batch size is always 256.

| Task ($\rightarrow$) Metric ($\rightarrow$) Dataset # ($\rightarrow$) | CLF Acc. 12 | Clust. V-Meas. 11 | PairCLF AP 3 | Rerank MAP 4 | Retrieval nDCG 15 | STS Spear. 10 | Summ. Spear. 1 | Avg. 56 |
|---|---|---|---|---|---|---|---|---|
| MEDI2 256 | 76.5 | 47.0 | 82.5 | 59.4 | 51.4 | 81.9 | 30.2 | 63.2 |
| MEDI2 4096 | 77.1 | 48.0 | 84.1 | 60.2 | 52.8 | 82.8 | 30.5 | 64.2 |
| Dataset ($\rightarrow$) Setup ($\rightarrow$) Metric ($\rightarrow$) | MMLU 0 FS EM | GSM8K 8 FS, CoT EM | BBH 3 FS, CoT EM | TyDi QA 1 FS, GP F1 | HumanEval 0 FS pass@1 | Alpaca 0 FS, 1.0 % Win | | Avg. |
| MEDI2 256 | 57.1 | 49.0 | 53.3 | 55.3 | 32.3 | 73.6 | | 53.4 |
| MEDI2 4096 | 57.7 | 48.0 | 53.2 | 54.5 | 32.0 | 74.3 | | 53.3 |

Table 19: **Precision ablations.** BF16 refers to bfloat16 mixed precision and FP32 to float32 precision.

| Task ($\rightarrow$) Metric ($\rightarrow$) Dataset # ($\rightarrow$) | CLF Acc. 12 | Clust. V-Meas. 11 | PairCLF AP 3 | Rerank MAP 4 | Retrieval nDCG 15 | STS Spear. 10 | Summ. Spear. 1 | Avg. 56 |
|---|---|---|---|---|---|---|---|---|
| BF16 | 79.7 | 50.2 | 87.6 | 60.2 | 56.5 | 83.4 | 30.8 | 66.5 |
| FP32 | 79.6 | 50.3 | 87.2 | 59.9 | 56.1 | 83.3 | 30.9 | 66.3 |
| Dataset ($\rightarrow$) Setup ($\rightarrow$) Metric ($\rightarrow$) | MMLU 0 FS EM | GSM8K 8 FS, CoT EM | BBH 3 FS, CoT EM | TyDi QA 1 FS, GP F1 | HumanEval 0 FS pass@1 | Alpaca 0 FS, 1.0 % Win | | Avg. |
| BF16 | 58.2 | 51.5 | 52.8 | 55.9 | 37.3 | 74.4 | | 55.0 |
| FP32 | 55.9 | 52.0 | 49.9 | 53.9 | 31.2 | 71.3 | | 52.4 |

Table 20: **In-batch negatives ablations.**

| Task ($\rightarrow$) Metric ($\rightarrow$) Dataset # ($\rightarrow$) | CLF Acc. 12 | Clust. V-Meas. 11 | PairCLF AP 3 | Rerank MAP 4 | Retrieval nDCG 15 | STS Spear. 10 | Summ. Spear. 1 | Avg. 56 |
|---|---|---|---|---|---|---|---|---|
| Any dataset | 79.7 | 49.8 | 85.5 | 59.8 | 54.9 | 83.9 | 30.5 | 66.0 |
| Same dataset | 79.5 | 48.9 | 87.4 | 59.0 | 56.2 | 83.0 | 30.5 | 66.0 |
| Dataset ($\rightarrow$) Setup ($\rightarrow$) Metric ($\rightarrow$) | MMLU 0 FS EM | GSM8K 8 FS, CoT EM | BBH 3 FS, CoT EM | TyDi QA 1 FS, GP F1 | HumanEval 0 FS pass@1 | Alpaca 0 FS, 1.0 % Win | | Avg. |
| Any dataset | 56.1 | 43.5 | 53.1 | 46.6 | 33.5 | 72.3 | | 50.9 |
| Same dataset | 55.0 | 45.0 | 54.4 | 49.3 | 29.6 | 73.4 | | 51.1 |

Table 21: **Generative format ablations.**

| Dataset ($\rightarrow$) Setup ($\rightarrow$) Metric ($\rightarrow$) | MMLU 0 FS EM | GSM8K 8 FS, CoT EM | BBH 3 FS, CoT EM | TyDi QA 1 FS, GP F1 | HumanEval 0 FS pass@1 | Alpaca 0 FS, 1.0 % Win | | Avg. |
|---|---|---|---|---|---|---|---|---|
| Tülu 2 format | 57.5 | 52.0 | 55.4 | 56.6 | 34.5 | 75.4 | | 55.2 |
| Zephyr $\beta$ format | 57.3 | 53.5 | 52.7 | 59.1 | 0.0 | 71.2 | | 49.0 |

Table 22: **Unified models max tokens ablations.** X:Y refers to "maximum tokens allowed for embedding documents during training":"maximum tokens allowed for queries and documents during embedding evaluation". The sequence of Cs and Bs refers to the attention mechanism for (from left to right): Emb instruction, Emb sample, where C=Causal, B=Bidirectional, and Emb=Embedding.

| Task (→) | CLF | Clust. | PairCLF | Rerank | Retrieval | STS | Summ. | Avg. |
|---|---|---|---|---|---|---|---|---|
| Metric (→) | Acc. | V-Meas. | AP | MAP | nDCG | Spear. | Spear. | |
| Dataset # (→) | 12 | 11 | 3 | 4 | 15 | 10 | 1 | 56 |
| MEDI 2048:512 | 77.9 | 47.9 | 81.5 | 59.0 | 49.4 | 80.3 | 29.4 | 62.8 |
| MEDI 2048:4096 | 77.9 | 47.9 | 81.5 | 59.0 | 49.4 | 80.2 | 31.3 | 62.8 |
| MEDI 4096:512 | 76.7 | 47.3 | 79.8 | 58.8 | 47.0 | 78.5 | 30.0 | 61.3 |
| MEDI 4096:4096 | 76.8 | 47.2 | 79.8 | 58.8 | 46.9 | 78.2 | 29.9 | 61.3 |
| MEDI2 BBCC 2048:512 | 77.0 | 48.7 | 86.0 | 61.0 | 53.6 | 83.0 | 29.1 | 64.7 |
| MEDI2 BBCC 512:512 | 76.9 | 47.6 | 85.5 | 61.0 | 52.8 | 82.3 | 28.8 | 64.1 |

| Dataset (→) | MMLU | GSM8K | BBH | TyDi QA | HumanEval | Alpaca | Avg. |
|---|---|---|---|---|---|---|---|
| Setup (→) | 0 FS | 8 FS, CoT | 3 FS, CoT | 1 FS, GP | 0 FS | 0 FS, 1.0 | |
| Metric (→) | EM | EM | EM | F1 | pass@1 | % Win | |
| MEDI 2048:512/4096 | 57.4 | 45.0 | 53.1 | 56.0 | 32.3 | 72.9 | 52.8 |
| MEDI 4096:512/4096 | 53.8 | 43.0 | 52.7 | 54.8 | 30.1 | - | - |
| MEDI2 BBCC 2048:512 | 57.0 | 50.5 | 53.8 | 54.7 | 32.3 | 74.7 | 53.8 |
| MEDI2 BBCC 512:512 | 56.9 | 46.5 | 53.1 | 52.6 | 31.2 | 72.8 | 52.2 |

Table 23: **Loss ablations.** E.g. Mix (32 → 8) corresponds to token level loss across 32 samples and then sample level loss across 8 sub-batches for a total batch size of 256. E.g. 2.4 refers to the loss ratio of the 1st step: $\mathcal{L}_{\mathrm{Emb}}/\mathcal{L}_{\mathrm{Gen}}$.

| Task (→) | CLF | Clust. | PairCLF | Rerank | Retrieval | STS | Summ. | Avg. |
|---|---|---|---|---|---|---|---|---|
| Metric (→) | Acc. | V-Meas. | AP | MAP | nDCG | Spear. | Spear. | |
| Dataset # (→) | 12 | 11 | 3 | 4 | 15 | 10 | 1 | 56 |
| E5S Token 2.4 | 79.5 | 50.1 | 86.5 | 60.0 | 55.6 | 83.2 | 30.3 | 66.1 |
| E5S Token 6.0 | 79.7 | 50.2 | 87.6 | 60.2 | 56.5 | 83.4 | 30.8 | 66.5 |
| E5S Mix (32 → 8) 4.1 | 79.4 | 50.5 | 87.2 | 60.5 | 57.4 | 83.4 | 30.4 | 66.7 |

| Dataset (→) | MMLU | GSM8K | BBH | TyDi QA | HumanEval | Alpaca | Avg. |
|---|---|---|---|---|---|---|---|
| Setup (→) | 0 FS | 8 FS, CoT | 3 FS, CoT | 1 FS, GP | 0 FS | 0 FS, 1.0 | |
| Metric (→) | EM | EM | EM | F1 | pass@1 | % Win | |
| E5S Token 2.4 | 57.9 | 48.5 | 53.5 | 56.5 | 35.2 | 75.0 | 54.4 |
| E5S Token 6.0 | 58.2 | 51.5 | 52.8 | 55.9 | 37.3 | 74.4 | 55.0 |
| E5S Mix (32 → 8) 4.1 | 57.6 | 57.0 | 54.8 | 55.4 | 32.8 | 74.8 | 55.4 |
| MEDI2 Mix (4 → 64) 11.7 | 57.0 | 48.0 | 53.7 | 55.0 | 35.8 | 67.6 | 52.9 |
| MEDI2 Mix (32 → 8) 10.2 | 57.0 | 50.5 | 53.8 | 54.7 | 32.3 | 74.7 | 53.8 |

## K    GRITLM MTEB FULL RESULTS

Table 24: **MTEB full results from Table 1.**

| Dataset | Gen-only | Emb-only | GRITLM 7B | 8X7B |
|---|---|---|---|---|
| AmazonCounterfactualClassification | 70.06 | 82.55 | 81.18 | 80.48 |
| AmazonPolarityClassification | 74.74 | 96.19 | 96.52 | 96.32 |
| AmazonReviewsClassification | 38.63 | 57.28 | 57.81 | 57.18 |
| Banking77Classification | 71.25 | 88.73 | 88.47 | 87.46 |
| EmotionClassification | 36.61 | 51.83 | 52.81 | 50.06 |
| ImdbClassification | 73.94 | 94.58 | 95.00 | 94.32 |
| MassiveIntentClassification | 66.82 | 79.37 | 80.78 | 79.72 |
| MassiveScenarioClassification | 71.27 | 81.20 | 82.09 | 81.09 |
| MTOPDomainClassification | 85.40 | 96.72 | 96.16 | 95.29 |
| MTOPIntentClassification | 75.60 | 87.19 | 87.13 | 87.08 |
| ToxicConversationsClassification | 66.36 | 68.37 | 70.80 | 70.89 |
| TweetSentimentExtractionClassification | 54.61 | 61.91 | 64.78 | 62.48 |
| ArxivClusteringP2P | 45.40 | 50.87 | 51.67 | 50.72 |
| ArxivClusteringS2S | 29.86 | 47.35 | 48.11 | 48.01 |
| BiorxivClusteringP2P | 33.45 | 40.18 | 40.87 | 41.41 |
| BiorxivClusteringS2S | 23.02 | 39.60 | 39.80 | 38.67 |
| MedrxivClusteringP2P | 27.49 | 36.61 | 36.52 | 36.54 |
| MedrxivClusteringS2S | 23.17 | 37.28 | 36.80 | 37.24 |
| RedditClustering | 23.28 | 63.52 | 61.30 | 63.01 |
| RedditClusteringP2P | 55.00 | 67.81 | 67.26 | 65.86 |
| StackExchangeClustering | 47.14 | 75.53 | 77.33 | 74.41 |
| StackExchangeClusteringP2P | 33.95 | 46.22 | 41.33 | 38.52 |
| TwentyNewsgroupsClustering | 18.15 | 56.8 | 55.70 | 57.16 |
| SprintDuplicateQuestions | 51.57 | 93.37 | 93.00 | 91.24 |
| TwitterSemEval2015 | 50.60 | 80.61 | 81.08 | 77.21 |
| TwitterURLCorpus | 60.36 | 87.20 | 87.40 | 86.45 |
| AskUbuntuDupQuestions | 49.02 | 68.13 | 67.34 | 65.60 |
| MindSmallReranking | 27.83 | 32.19 | 31.81 | 32.84 |
| SciDocsRR | 56.65 | 87.00 | 86.84 | 86.43 |
| StackOverflowDupQuestions | 38.42 | 55.48 | 55.96 | 54.33 |
| ArguAna | 35.96 | 62.95 | 63.24 | 59.49 |
| ClimateFEVER | 8.96 | 31.09 | 30.91 | 28.69 |
| CQADupstackRetrieval | 7.20 | 50.83 | 49.42 | 47.63 |
| DBPedia | 2.15 | 47.06 | 46.60 | 46.54 |
| FEVER | 5.02 | 85.41 | 82.74 | 85.02 |
| FiQA2018 | 6.27 | 60.22 | 59.95 | 49.89 |
| HotpotQA | 6.67 | 79.15 | 79.40 | 73.83 |
| MSMARCO | 0.66 | 41.55 | 41.96 | 35.55 |
| NFCorpus | 3.74 | 41.69 | 40.89 | 39.05 |
| NQ | 2.14 | 69.46 | 70.30 | 63.87 |
| QuoraRetrieval | 64.42 | 89.08 | 89.47 | 87.70 |
| SCIDOCS | 2.32 | 24.86 | 24.41 | 23.06 |
| SciFact | 35.58 | 78.92 | 79.17 | 77.02 |
| Touche2020 | 3.06 | 24.30 | 27.93 | 27.97 |
| TRECCOVID | 20.92 | 75.29 | 74.8 | 81.07 |
| BIOSSES | 70.87 | 86.20 | 86.35 | 87.34 |
| SICK-R | 58.95 | 83.03 | 83.13 | 80.56 |
| STS12 | 44.25 | 78.07 | 77.34 | 73.69 |
| STS13 | 64.22 | 85.98 | 85.04 | 85.82 |

| | | | | |
|---|---|---|---|---|
| STS14 | 52.24 | 83.92 | 82.91 | 82.05 |
| STS15 | 64.53 | 89.18 | 88.13 | 88.8 |
| STS16 | 65.89 | 86.83 | 86.24 | 86.2 |
| STS17 | 69.64 | 89.7 | 90.13 | 91.46 |
| STS22 | 57.29 | 68.41 | 68.63 | 69.21 |
| STSBenchmark | 53.89 | 86.74 | 85.64 | 87.43 |
| SummEval | 21.14 | 30.18 | 30.37 | 29.82 |
| **Average** | **41.21** | **66.82** | **66.76** | **65.66** |

## L    REDUCING EMBEDDING TRAINING MEMORY

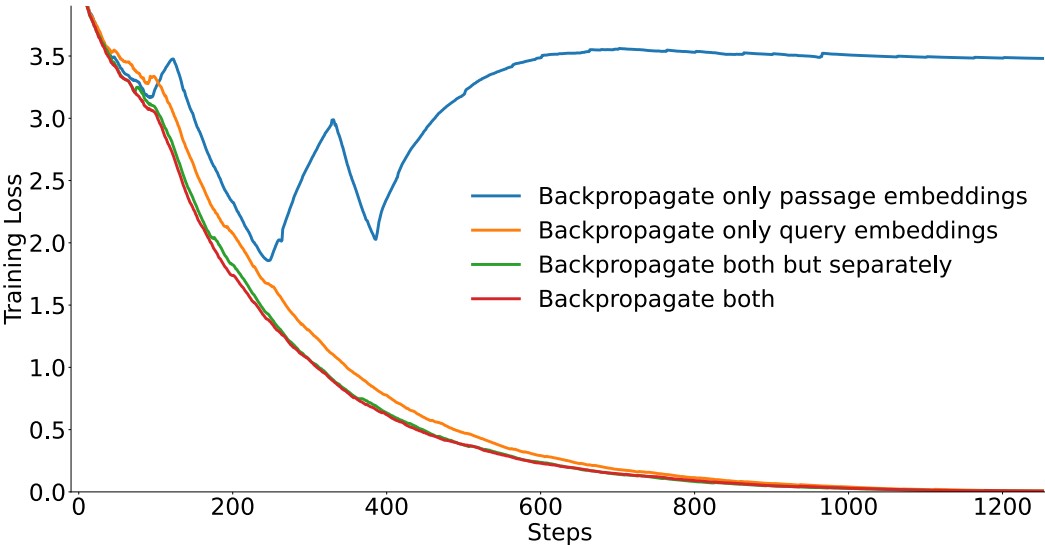

Figure 9: **Embedding memory ablations.** Passage corresponds to both positive and document embeddings. Loss is smoothed with exponential moving average smoothing and a weight of 0.99.

Generative training only requires sufficient memory to perform a forward and backward pass on a single training sample of a given sequence length. Meanwhile, naive embedding training with in-batch negatives requires sufficient memory to accommodate a forward and a backward pass on $3 * bs$ samples. The 3 corresponds to the need for passing a triplet of a query, a positive, and a negative document (Equation 1). The batch size ($bs$) factor corresponds to the need for forwarding all samples together as regular gradient accumulation does not work with in-batch negatives. Below we outline the strategies we employ to reduce these memory needs.

**Triplet**    As the full triplet is only required for loss calculation (Equation 1), it can be split across separate forward and backward passes. To avoid the memory requirements of gradients in PyTorch Autograd (Paszke et al., 2019), this requires two additional forward passes without gradients. Simplified code representing this procedure is depicted in Listing 1. In our training, it was sufficient to only split the triplet into two parts: query and passages, where passages consist of both a positive and a negative document. Thus, we only incur the cost of one additional forward pass without gradients on the query. Alternatively, one could only backpropagate on a subset of the embeddings, however, we show in Figure 9 that this leads to worse performance.

**In-batch negatives**    There are two strategies to reduce the batch size memory requirement to that of a single batch while using nearly unlimited in-batch negatives. **(1) Distributed Training:** The best strategy is to distribute the training across up to $bs$ GPUs. The representations can then be gathered across GPUs to compute the contrastive loss with in-batch negatives. **(2) GradCache:** If enough GPUs are not available, GradCache (Gao et al., 2021b) can be used. GradCache maintains

Listing 1: **Splitting of the embedding pass to save memory, simplified.**

```python
def distributed_contrastive_loss(q, p, n):
    # Gather in-batch negatives across devices...
    # Compute contrastive loss...

# Split triplet into three forward passes
pos_rep = model(pos)
with torch.no_grad():
    q_rep = model(query)
    neg_rep = model(neg)

# Only perform backward pass on positive documents
loss = distributed_contrastive_loss(q_rep, pos_rep, neg_rep)
loss.backward()

pos_rep = pos_rep.detach()
# Perform forward + backward on negatives & reuse rest
neg_rep = model(neg)
loss = distributed_contrastive_loss(q_rep, pos_rep, neg_rep)
loss.backward()

# Perform forward + backward on queries & reuse rest
neg_rep = neg_rep.detach()
q_rep = model(query)
loss = distributed_contrastive_loss(q_rep, pos_rep, neg_rep)
loss.backward()
```

in-batch negatives while allowing computation of gradients for each triplet at a time, thus effectively corresponding to gradient accumulation for contrastive loss. However, it comes at the cost of additional forward passes.

Across training runs, we make use of all three strategies (splitting, distributed training, GradCache).

## M  HYPERPARAMETERS

We finetune all parameters of our models for up to 1253 steps. Our learning rate is 2e-5, we use 3% of steps for linear warm-up of the learning rate and decay it linearly to 0 over training. To save memory, we use PyTorch FSDP (Zhao et al., 2023), gradient checkpointing, BF16 mixed precision training, and strategies outlined in Appendix L. During training, we use a sequence length of 2048 for generative samples, 256 for embedding queries, and 2048 for embedding documents unless otherwise specified. We finetune using the Adam optimizer (Kingma & Ba, 2017) with beta1=0.9 and beta2=0.999 and no weight decay. We also use Flash-Attention 2 (Dao et al., 2022; Dao, 2023) via PyTorch SDPA.

We evaluate models using the settings put forth by the creators of MTEB (Muennighoff et al., 2023c), Tülu (Ivison et al., 2023; Wang et al., 2024) and HumanEvalSynthesize (Muennighoff et al., 2023a; Zhuo et al., 2024). For MTEB, we evaluate using a maximum sequence length of 512 unless otherwise specified.

## N  EMBEDDING INSTRUCTION FOR GENERATIVE MODELS

As prior instruction-tuned models have been trained without an embedding objective, it is unclear whether one should add an instruction when evaluating them on embedding tasks. We benchmark the Mistral 7B instruct model on MTEB with and without instruction in Table 25. We find that performance is around the same, however, adding instructions performs slightly better. Thus, we add an instruction for all instruction-tuned models when benchmarking their embedding performance.

Table 25: **Benchmarking the benefit of an embedding instruction for generative instruction-tuned models.** When an instruction is used ("Mistral Instruct w/"), we use the default instructions from Instructor XL with the prompt template of the Mistral Instruct model. For no instruction ("Mistral Instruct w/o"), the procedure is the same as for the base model ("Mistral")

| Task (→) | CLF | Clust. | PairCLF | Rerank | Retrieval | STS | Summ. | Avg. |
|---|---|---|---|---|---|---|---|---|
| Metric (→) | Acc. | V-Meas. | AP | MAP | nDCG | Spear. | Spear. | |
| Dataset # (→) | 12 | 11 | 3 | 4 | 15 | 10 | 1 | 56 |
| Mistral | 63.5 | 34.6 | 53.5 | 43.2 | 13.2 | 57.4 | 19.7 | 40.5 |
| Mistral Instruct w/o | 65.4 | 35.6 | 60.2 | 44.6 | 16.8 | 61.1 | 25.9 | 43.3 |
| Mistral Instruct w/ | 67.1 | 34.6 | 59.6 | 44.8 | 16.3 | 63.4 | 25.9 | 43.7 |

## O HUMANEVAL FORMAT

In Tülu 2 (Ivison et al., 2023), models are evaluated on HumanEval (Chen et al., 2021) without the model's chat format. As this does not reflect the intended usage of the models, we instead use the appropriate chat format for evaluating HumanEval. To do so, we use the instructions and evaluation procedure from HumanEval-Synthesize (Muennighoff et al., 2023a). In Table 26 we benchmark the impact this has on performance for the Tülu 2 7B model (Ivison et al., 2023). We find that the performance is around equivalent and thus use the chat format for all evaluations of chat models. For non-chat models, we use the original HumanEval continuation format as proposed by Chen et al. (2021)

Table 26: **HumanEvalSynthesize with different formats using Tülu 2 7B.**

| Format | Tülu 2 7B | |
|---|---|---|
| | No Chat | Chat |
| Pass@1 | 23.4 | 24.5 |
| Pass@10 | 32.4 | 31.3 |

## P EMBEDDING IN FP32 VS BF16

We perform all training and evaluations in BF16 (bfloat16) mixed precision to speed up computations. We verified that it performs comparably to FP32 (float32) on MTEB in Table 27. Note that pooling and subsequent similarity computations are still in FP32.

Table 27: **Embeddings in FP32 vs BF16.** Benchmarking of the raw Mistral 7B model. "FP32" corresponds to doing all computations in float32 precision. "BF16" and "BF16 Cache" corresponds to doing most operations in bfloat16 except for operations that PyTorch auto casts to float32 (e.g. normalization), pooling and similarity computations. For "BF16 Cache", we cast the embeddings after pooling to BF16 and then back to FP32 before similarity computations. This corresponds to locally caching the embeddings in BF16 to save storage and then casting them to FP32 at inference.

| Task (→) | CLF | Clust. | PairCLF | Rerank | Retrieval | STS | Summ. | Avg. |
|---|---|---|---|---|---|---|---|---|
| Metric (→) | Acc. | V-Meas. | AP | MAP | nDCG | Spear. | Spear. | |
| Dataset # (→) | 12 | 11 | 3 | 4 | 15 | 10 | 1 | 56 |
| FP32 | 63.46 | 34.62 | 53.56 | 43.24 | 13.26 | 57.38 | 19.87 | 40.51 |
| BF16 | 63.47 | 34.60 | 53.52 | 43.24 | 13.24 | 57.38 | 19.68 | 40.50 |
| BF16 Cache | 63.47 | 34.56 | 53.52 | 43.25 | 13.11 | 57.38 | 19.71 | 40.46 |

## Q   Unreliability of MT-Bench

We experiment with using MT-Bench with its rec-
ommended absolute scores for our generative eval-
uation (Zheng et al., 2023). However, we find that as
soon as we switch the LLM Evaluator from GPT-4
to GPT-4 Turbo, the scores change significantly (Ta-
ble 28). GPT-4 is a closed-source model with changes
happening behind the scenes that users may not know
about (Chen et al., 2023). Thus, if OpenAI decides to
change GPT-4, all existing MT-Bench absolute scores
would essentially become obsolete. The same applies
if the API is retired. To alleviate this, we also exper-
iment with using Zephyr 7B $\beta$ (Tunstall et al., 2023)
and Llama 2 70B Chat (Touvron et al., 2023) as eval-

Table 28: **Using GPT-4 vs GPT-4 Turbo as
a judge for MT-Bench.** Each evaluator is
provided with the same generations of the
same instruction-tuned model.

|        | GPT-4 | GPT-4 Turbo | Drop |
|--------|-------|-------------|------|
| Turn 1 | 4.08  | 3.05        | 25%  |
| Turn 2 | 2.64  | 1.88        | 29%  |
| Avg.   | 3.36  | 2.48        | 26%  |

uators, however, we find them to often not provide any rating as they struggle to understand the
prompt. While AlpacaEval (Dubois et al., 2023; Li et al., 2023b), which we use, shares some of
these problems, its comparison-based evaluation is more stable. This is because comparing if one
generation is better than another generally has an objective ground truth solution. Meanwhile, there
is no objective solution as to whether an absolute score of a given generation should be 3 or 4
(MT-Bench has eleven levels from 0-10). This is up to the subjective value system of the evaluator.

## R   Limitations and Future Work

**Efficiency**   As mentioned in §1, training using GRIT requires more compute than only embedding or
only generative training as two forward and backward passes are required. As finetuning is generally
cheaper than pretraining, this is not a major problem, but efficiency improvements would nonetheless
be worthwhile. One potential way to improve efficiency would be to extract the embedding and
generative signal from the same samples, rather than separate samples. This could halve the number
of forward passes required, yet due to the different loss functions, it may not make the backward
passes significantly faster.

**Performance improvements**   While we find that GRITLM performs strongly on embedding and
generative tasks (§3.2), there have been many recent models with even stronger performance in
either embedding or generative tasks; yet not the combination of both. A natural future work would
therefore be extending the GRIT approach to more recent models, such as the Llama-3 series of
models (Dubey et al., 2024) to build stronger models that can handle both embedding and generation.

**Caching improvements**   As we outline in §5, the caching variants with GRITLM suffer from
attention mismatch problems. Further, doc caching requires a significant amount of extra storage.
While storage is usually cheap, it may nonetheless be prohibitively expensive for very large indices.
One promising avenue for future work is improving caching with GRITLM, such as via finetuning
with caching, such that it learns to deal with the mismatch problem.

**GRITLM Agents**   Future work may consider using the embedding capability to let the generative
model initiate a search over an index when it deems necessary. Currently, this is often accomplished
via external retrieval plugins. Such plugins are no longer necessary if the model can retrieve on
its own. Teaching the model to invoke its own embedding capability likely requires additional
finetuning (just like teaching it to invoke an external plugin (Schick et al., 2023)). A sample could
look something like:

```
"<|user|>\nWhat is the capital of Japan?\n<|internal|>\nI am not
sure I know this. Let me produce an embedding for it and search
for the answer. Retrieve answers for this query.\n<|embed|>\nWhat
is the capital of Japan?\n<|output|>\nTokyo, Japan's busy capital,
mixes the ultramodern and the traditional..\n<|assistant|>\n
The capital of Japan is Tokyo.\n"
```

**Pretraining**   For our experiments we take an off-the-shelf pretrained language model. However, it should also be possible to use the GRIT approach to pretrain from scratch. As labeled embedding data is likely too scarce for pretraining, one could either rely on unsupervised approaches for the embedding objective, such as RetroMAE (Xiao et al., 2022; Xiao & Liu, 2022), or use methods like data augmentation (Dhole et al., 2022), pruning (Xia et al., 2023) or multi-epoch training to deal with the data constraint (Muennighoff et al., 2023b; Luukkonen et al., 2023).

**Format Efficiency**   Our format in Figure 3 is inefficient, as encoding the embedding format, `<|user|>\n<|embed|>\n`, requires 13 tokens and encoding the generative format, `<|user|>\n<|assistant|>\n`, requires 15 tokens. Using special tokens could simplify this and thus make training and inference slightly cheaper.

**Training efficiency: Packing and Reusing**   It is common to pack samples during generative instruction tuning to maximize efficiency (Chung et al., 2022; Muennighoff et al., 2023d). Packing embedding samples during training should also be possible by ensuring attention is only paid to each respective sample. Going even further is it possible to pack generative and embedding training data into the same sample and reuse the same sample for both tasks? This could look similar to the example provided in "GRITLM Agents" with the generative loss applied over the assistant response and the contrastive loss applied to the representation of the text following "`<|embed|>`". By reusing samples it may be possible to significantly decrease the resources needed for GRIT.

## S   DATASET COMPOSITION

Table 29: **E5S dataset composition.**

| Dataset (↓) | Num samples |
|---|---|
| DuReader (Qiu et al., 2022) | 86,395 |
| ELI5 (Fan et al., 2019) | 50293 |
| FEVER (Thorne et al., 2018) | 71,257 |
| GPT4 Bitext (Wang et al., 2024) | 89,324 |
| GPT4 P2P (Wang et al., 2024) | 16,842 |
| GPT4 P2S (Wang et al., 2024) | 121,878 |
| GPT4 Retrieval (Wang et al., 2024) | 166,602 |
| GPT4 S2S (Wang et al., 2024) | 13,481 |
| GPT4 STS (Wang et al., 2024) | 98,626 |
| HotpotQA (Yang et al., 2018) | 68,659 |
| NLI (Gao et al., 2022) | 275,601 |
| MIRACL (Zhang et al., 2022) | 40,203 |
| MSMARCO (Bajaj et al., 2018) | 244,582 |
| MSMARCO Doc (Bajaj et al., 2018) | 71,594 |
| Mr. TyDi (Zhang et al., 2021) | 48,729 |
| NQ (Kwiatkowski et al., 2019) | 71,408 |
| S2ORC (Lo et al., 2020) | 80,000 |
| SQuAD (Rajpurkar et al., 2016) | 87,599 |
| T2Ranking (Xie et al., 2023) | 112,335 |
| TriviaQA (Karpukhin et al., 2020) | 60,296 |
| Quora (DataCanary et al., 2017) | 14,926 |
| Total | 1,890,630 |

Table 30: **MEDI2 dataset composition.**

| MEDI Dataset (↓) | Num samples |
|---|---|
| AGNews (Zhang et al., 2016) | 199,792 |
| Altlex (Hidey & McKeown, 2016) | 112,602 |
| Amazon QA (Gupta et al., 2019) | 199,180 |
| Amazon Review (Keung et al., 2020) | 198,298 |
| CC News (Hamborg et al., 2017) | 190,503 |
| CNN/Dailymail (Fabbri et al., 2021) | 189,407 |
| COCO Captions (Chen et al., 2015) | 82,783 |
| ELI5 (Fan et al., 2019) | 196,572 |
| FEVER KILT (Thorne et al., 2018; Petroni et al., 2021) | 71,257 |
| Flickr 30k (Young et al., 2014) | 31,783 |
| Gigaword (Rush et al., 2015; Graff et al., 2003) | 200,000 |
| GooAQ (Khashabi et al., 2021) | 199,981 |
| HotpotQA KILT (Yang et al., 2018; Petroni et al., 2021) | 65,351 |
| NLI (Gao et al., 2022) | 277,195 |
| MSMARCO (Bajaj et al., 2018) | 491,980 |
| MedMCQA (Pal et al., 2022) | 156,905 |
| Multi-LexSum (Shen et al., 2022) | 2,771 |
| NPR (Team, 2021b) | 193,399 |
| NQ (Kwiatkowski et al., 2019) | 73,226 |
| PAQ (Lewis et al., 2021b) | 190,162 |
| PubMedQA (Jin et al., 2019) | 190,481 |
| Reddit (Team, 2021c) | 196,247 |
| S2ORC (Lo et al., 2020) | 193,458 |
| SQuAD (Rajpurkar et al., 2016) | 84,105 |
| SciTLDR (Cachola et al., 2020) | 1,742 |
| SearchQA (Dunn et al., 2017) | 114,520 |
| Sentence Compression (Filippova & Altun, 2013) | 179,996 |
| SimpleWiki (Coster & Kauchak, 2011) | 102,035 |
| StackExchange (Team, 2021a) | 201,050 |
| SuperNI (300 datasets) (Wang et al., 2022c) | 2,682,465 |
| SPECTER (Cohan et al., 2020) | 684,000 |
| T-REx KILT (ElSahar et al., 2018; Petroni et al., 2021) | 191,383 |
| Quora (DataCanary et al., 2017) | 101,762 |
| WikiAnswers (Fader et al., 2014) | 200,000 |
| WikiHow (Koupaee & Wang, 2018) | 128,542 |
| XSum (Narayan et al., 2018) | 190,427 |
| Yahoo (Zhang et al., 2016) | 198,346 |
| Zeroshot KILT (Levy et al., 2017; Petroni et al., 2021) | 124,547 |
| Total | 9,084,806 |

# T    DATASET SAMPLES

---

**Query instruction:**

---

Represent the sentence for retrieving supporting documents;

---

**Query sample:**

---

what two plates form the san andreas fault

---

**Positive instruction:**

---

Represent the document for retrieval;

---

**Positive sample:**

---

The San Andreas Fault marks the junction between the North American and Pacific Plates. The fault is 1300 km long, extends to at least 25 km in depth, and has a north west-south east trend. It is classified as a right lateral (dextral) strike-slip fault. Loading the player ...

---

**Negative instruction:**

---

Represent the document for retrieval;

---

**Negative sample:**

---

The San Andreas Fault is the sliding boundary between the Pacific Plate and the North American Plate. It slices California in two from Cape Mendocino to the Mexican border. San Diego, Los Angeles and Big Sur are on the Pacific Plate.

---

Figure 10: **MEDI sample.**

---

**Query instruction:**

---

Represent this question to retrieve a fitting Wikipedia passage (formal)

---

**Query sample:**

---

which two plates meet along the west coast of the USA

---

**Positive instruction:**

---

Represent this Wikipedia text in order to get a user query which it answers!

---

**Positive sample:**

---

on to a transitional deformation zone in the Chersky Range, then the Ulakhan Fault between it and
the Okhotsk Plate, and finally the Aleutian Trench to the end of the Queen Charlotte Fault system.
The westerly boundary is the Queen Charlotte Fault running offshore along the coast of Alaska and
the Cascadia subduction zone to the north, the San Andreas Fault through California, the East Pacific
Rise in the Gulf of California, and the Middle America Trench to the south.
On its western edge, the Farallon Plate has been subducting

---

**Negative instruction:**

---

Represent this passage to easily find a natural-written user question that can be answered by it.

---

**Negative sample:**

---

the continental margin.
Types.
There are two types of continental margins: ä̈ctiveänd p̈assivem̈argins.
Active margins are typically associated with lithospheric plate boundaries. These active margins can
be convergent or transform margins, and are also places of high tectonic activity, including volcanoes
and earthquakes. The West Coast of North America and South America are active margins. Active
continental margins are typically narrow from coast to shelf break, with steep descents into trenches.
Convergent active margins occur where oceanic plates meet continental

Figure 11: **MEDI2 sample.**

**Query instruction:**

Given a question, retrieve Wikipedia passages that answer the question

**Query sample:**

which two plates meet along the west coast of the USA

**Positive sample:**

North American Plate boundary is the Queen Charlotte Fault running offshore along the coast of Alaska and the Cascadia subduction zone to the north, the San Andreas Fault through California, the East Pacific Rise in the Gulf of California, and the Middle America Trench to the south. On its western edge, the Farallon Plate has been subducting under the North American Plate since the Jurassic Period. The Farallon Plate has almost completely subducted beneath the western portion of the North American Plate leaving that part of the North American Plate in contact with the Pacific Plate as the San Andreas Fault. The Juan

**Negative sample:**

Caribbean Plate Caribbean Plate The Caribbean Plate is a mostly oceanic tectonic plate underlying Central America and the Caribbean Sea off the north coast of South America. Roughly 3.2 million square kilometers (1.2 million square miles) in area, the Caribbean Plate borders the North American Plate, the South American Plate, the Nazca Plate and the Cocos Plate. These borders are regions of intense seismic activity, including frequent earthquakes, occasional tsunamis, and volcanic eruptions. The northern boundary with the North American plate is a transform or strike-slip boundary which runs from the border area of Belize, Guatemala (Motagua Fault), and Honduras in Central

Figure 12: **E5 sample.** The E5 dataset does not use instructions for documents, thus the positive and negative samples do not have instructions.

**Instruction:**

Q: Lloyd, Mark, and Michael have their Pokemon cards collection. Currently, Mark has thrice as many cards as Lloyd but has 10 fewer cards than Michael. If Michael has 100 cards now, how many more cards should they collect so that all three of them will have a total of 300 cards?
A: 80
Explain how we arrive at this answer:

**Sample:**

Explanation: Mark has 10 fewer cards than Michael so Mark has 100 cards - 10 cards = 90 cards. So, Lloyd has 90 cards / 3 = 30 cards. All three of them have 90 cards + 30 cards + 100 cards = 220 cards. Thus, they need to collect 300 cards - 220 cards = 80 more cards.

Figure 13: **Tülu 2 sample.**

## U   EVALUATION PROMPTS

### U.1   EMBEDDING PROMPTS

Table 31 contains the prompt for each MTEB dataset when training on the E5 dataset, which are the same instructions as used in Wang et al. (2024). Table 32 contains the MTEB prompts we use when training on MEDI2, which we wrote ourselves. For models trained on MEDI, we use the instructions for Instructor-XL from Su et al. (2023).

Table 31: **Instructions used for evaluation on the MTEB benchmark when training with the E5 dataset.** "STS*" indicates we use the same instructions for all the STS tasks. For retrieval datasets, we do not use an instruction for the document and only display the query instruction.

| Task Name | Instruction |
|---|---|
| AmazonCounterfactualClassif. | Classify a given Amazon customer review text as either counterfactual or not-counterfactual |
| AmazonPolarityClassification | Classify Amazon reviews into positive or negative sentiment |
| AmazonReviewsClassification | Classify the given Amazon review into its appropriate rating category |
| Banking77Classification | Given a online banking query, find the corresponding intents |
| EmotionClassification | Classify the emotion expressed in the given Twitter message into one of the six emotions: anger, fear, joy, love, sadness, and surprise |
| ImdbClassification | Classify the sentiment expressed in the given movie review text from the IMDB dataset |
| MassiveIntentClassification | Given a user utterance as query, find the user intents |
| MassiveScenarioClassification | Given a user utterance as query, find the user scenarios |
| MTOPDomainClassification | Classify the intent domain of the given utterance in task-oriented conversation |
| MTOPIntentClassification | Classify the intent of the given utterance in task-oriented conversation |
| ToxicConversationsClassif. | Classify the given comments as either toxic or not toxic |
| TweetSentimentClassification | Classify the sentiment of a given tweet as either positive, negative, or neutral |
| ArxivClusteringP2P | Identify the main and secondary category of Arxiv papers based on the titles and abstracts |
| ArxivClusteringS2S | Identify the main and secondary category of Arxiv papers based on the titles |
| BiorxivClusteringP2P | Identify the main category of Biorxiv papers based on the titles and abstracts |
| BiorxivClusteringS2S | Identify the main category of Biorxiv papers based on the titles |
| MedrxivClusteringP2P | Identify the main category of Medrxiv papers based on the titles and abstracts |
| MedrxivClusteringS2S | Identify the main category of Medrxiv papers based on the titles |
| RedditClustering | Identify the topic or theme of Reddit posts based on the titles |
| RedditClusteringP2P | Identify the topic or theme of Reddit posts based on the titles and posts |

| | |
|---|---|
| StackExchangeClustering | Identify the topic or theme of StackExchange posts based on the titles |
| StackExchangeClusteringP2P | Identify the topic or theme of StackExchange posts based on the given paragraphs |
| TwentyNewsgroupsClustering | Identify the topic or theme of the given news articles |
| SprintDuplicateQuestions | Retrieve duplicate questions from Sprint forum |
| TwitterSemEval2015 | Retrieve tweets that are semantically similar to the given tweet |
| TwitterURLCorpus | Retrieve tweets that are semantically similar to the given tweet |
| AskUbuntuDupQuestions | Retrieve duplicate questions from AskUbuntu forum |
| MindSmallReranking | Retrieve relevant news articles based on user browsing history |
| SciDocsRR | Given a title of a scientific paper, retrieve the titles of other relevant papers |
| StackOverflowDupQuestions | Retrieve duplicate questions from StackOverflow forum |
| ArguAna | Given a claim, find documents that refute the claim |
| ClimateFEVER | Given a claim about climate change, retrieve documents that support or refute the claim |
| CQADupstackRetrieval | Given a question, retrieve detailed question descriptions from Stackexchange that are duplicates to the given question |
| DBPedia | Given a query, retrieve relevant entity descriptions from DBPedia |
| FEVER | Given a claim, retrieve documents that support or refute the claim |
| FiQA2018 | Given a financial question, retrieve user replies that best answer the question |
| HotpotQA | Given a multi-hop question, retrieve documents that can help answer the question |
| MSMARCO | Given a web search query, retrieve relevant passages that answer the query |
| NFCorpus | Given a question, retrieve relevant documents that best answer the question |
| NQ | Given a question, retrieve Wikipedia passages that answer the question |
| QuoraRetrieval | Given a question, retrieve questions that are semantically equivalent to the given question |
| SCIDOCS | Given a scientific paper title, retrieve paper abstracts that are cited by the given paper |
| SciFact | Given a scientific claim, retrieve documents that support or refute the claim |
| Touche2020 | Given a question, retrieve detailed and persuasive arguments that answer the question |
| TRECCOVID | Given a query on COVID-19, retrieve documents that answer the query |
| STS* | Retrieve semantically similar text. |
| SummEval | Given a news summary, retrieve other semantically similar summaries |

Table 32: **Instructions used for evaluation on the MTEB benchmark when training with the MEDI2 dataset.** For asymmetric datasets, Q refers to instructions for queries, while D refers to document instructions.

| Task Name | Instruction |
|---|---|
| AmazonCounterfactualClassification | Represent the text to find another sentence with the same counterfactuality, e.g. sentences with "would", "wish", etc. should match with other sentences of that kind. |
| AmazonPolarityClassification | Represent the review for finding another Amazon review with the same sentiment (positive / negative) |
| AmazonReviewsClassification | Represent the review for finding another Amazon review with the same rating |
| Banking77Classification | Represent the text for finding another one-sentence banking query with the same intent |
| EmotionClassification | Represent the text for finding another one-sentence text with the same emotion |
| ImdbClassification | Represent the text for finding another one-sentence movie review with the same sentiment |
| MassiveIntentClassification | Represent the text for finding another text of a few words with the same intent |
| MassiveScenarioClassification | Represent the text for finding another text of a few words about the same scenario |
| MTOPDomainClassification | Represent the text for finding another text of a few words about the same domain |
| MTOPIntentClassification | Represent the text for finding another text of a few words with the same intent |
| ToxicConversationsClassification | Represent the text for finding another comment of up to a passage in length with the same level of toxicity (either toxic or not toxic) |
| TweetSentimentExtractionClassification | Represent the tweet for finding another tweet with the same sentiment (positive / neutral / negative) |
| ArxivClusteringP2P | Represent the text to find another arXiv title with abstract (concatenated) about the same topic |
| ArxivClusteringS2S | Represent the text to find another arXiv title about the same topic |
| BiorxivClusteringP2P | Represent the text to find another bioRxiv title with abstract (concatenated) about the same topic |
| BiorxivClusteringS2S | Represent the text to find another bioRxiv title about the same topic |
| MedrxivClusteringS2S | Represent the text to find another medRxiv title about the same topic |
| MedrxivClusteringP2P | Represent the text to find another medRxiv title with abstract (concatenated) about the same topic |
| RedditClustering | Represent the text to find another Reddit community title that stems from the same subreddit |
| RedditClusteringP2P | Represent the text to find another Reddit community title with post (concatenated) from the same subreddit |
| StackExchangeClustering | Represent the text to find another StackExchange title that stems from the same StackExchange |

| | |
|---|---|
| StackExchangeClusteringP2P | Represent the text to find another StackExchange title with post (concatenated) that stems from the same StackExchange |
| TwentyNewsgroupsClustering | Represent the title to find a similar news title from the same newsgroup |
| SprintDuplicateQuestions | Represent the question to be matched with another duplicate user question from the Sprint community forum |
| TwitterSemEval2015 | Represent the tweet to find another tweet that is a paraphrase of it |
| TwitterURLCorpus | Represent the tweet to find another tweet that is a paraphrase of it |
| ArguAna Q | Represent the passage to find a passage with a counter-argument about the same topic to it |
| ArguAna D | Represent the passage to find a passage with a counter-argument about the same topic to it |
| ClimateFEVER Q | Represent the climate-based claim to find a Wikipedia abstract to support it |
| ClimateFEVER D | Represent the Wikipedia abstract to find a climate-related claim that it supports |
| CQADupstackAndroidRetrieval Q | Represent the title of a user question to find a duplicate user question title with body from the Android StackExchange forum |
| CQADupstackAndroidRetrieval D | Represent the question title with body posted by a user to find a duplicate user question title from the Android StackExchange forum |
| CQADupstackEnglishRetrieval Q | Represent the title of a user question to find a duplicate user question title with body from the English StackExchange forum |
| CQADupstackEnglishRetrieval D | Represent the question title with body posted by a user to find a duplicate user question title from the English StackExchange forum |
| CQADupstackGamingRetrieval Q | Represent the title of a user question to find a duplicate user question title with body from the Gaming StackExchange forum |
| CQADupstackGamingRetrieval D | Represent the question title with body posted by a user to find a duplicate user question title from the Gaming StackExchange forum |
| CQADupstackGisRetrieval Q | Represent the title of a user question to find a duplicate user question title with body from the Gis StackExchange forum |
| CQADupstackGisRetrieval D | Represent the question title with body posted by a user to find a duplicate user question title from the Gis StackExchange forum |
| CQADupstackMathematicaRetrieval Q | Represent the title of a user question to find a duplicate user question title with body from the Mathematica StackExchange forum |
| CQADupstackMathematicaRetrieval D | Represent the question title with body posted by a user to find a duplicate user question title from the Mathematica StackExchange forum |
| CQADupstackPhysicsRetrieval Q | Represent the title of a user question to find a duplicate user question title with body from the Physics StackExchange forum |
| CQADupstackPhysicsRetrieval D | Represent the question title with body posted by a user to find a duplicate user question title from the Physics StackExchange forum |

| | |
|---|---|
| CQADupstackProgrammersRetrieval Q | Represent the title of a user question to find a duplicate user question title with body from the Programmers StackExchange forum |
| CQADupstackProgrammersRetrieval D | Represent the question title with body posted by a user to find a duplicate user question title from the Programmers StackExchange forum |
| CQADupstackStatsRetrieval Q | Represent the title of a user question to find a duplicate user question title with body from the Stats StackExchange forum |
| CQADupstackStatsRetrieval D | Represent the question title with body posted by a user to find a duplicate user question title from the Stats StackExchange forum |
| CQADupstackTexRetrieval Q | Represent the title of a user question to find a duplicate user question title with body from the Tex StackExchange forum |
| CQADupstackTexRetrieval D | Represent the question title with body posted by a user to find a duplicate user question title from the Tex StackExchange forum |
| CQADupstackUnixRetrieval Q | Represent the title of a user question to find a duplicate user question title with body from the Unix StackExchange forum |
| CQADupstackUnixRetrieval D | Represent the question title with body posted by a user to find a duplicate user question title from the Unix StackExchange forum |
| CQADupstackWebmastersRetrieval Q | Represent the title of a user question to find a duplicate user question title with body from the Webmasters StackExchange forum |
| CQADupstackWebmastersRetrieval D | Represent the question title with body posted by a user to find a duplicate user question title from the Webmasters StackExchange forum |
| CQADupstackWordpressRetrieval Q | Represent the title of a user question to find a duplicate user question title with body from the Wordpress StackExchange forum |
| CQADupstackWordpressRetrieval D | Represent the question title with body posted by a user to find a duplicate user question title from the Wordpress StackExchange forum |
| DBPedia Q | Represent the entity to find a title with abstract about this entity from the DBPedia corpus |
| DBPedia D | Represent the title with abstract of a DBPedia corpus entry to find the entity of a few words it is about |
| FEVER Q | Represent the claim to find a Wikipedia abstract to support it |
| FEVER D | Represent the Wikipedia abstract to find a claim that it supports |
| FiQA2018 Q | Represent the StackExchange user query to find a StackExchange post from the Investment topic that answers it |
| FiQA2018 D | Represent the StackExchange post from the Investment topic to find a StackExchange user query that it answers |
| HotpotQA Q | Represent the multi-hop question to find a Wikipedia passage that answers it |
| HotpotQA D | Represent the Wikipedia passage to find a multi-hop question that it answers |
| MSMARCO Q | Represent the Bing user search query to find a passage that adequately addresses it |
| MSMARCO D | Represent the passage for finding a Bing user search query about it |
| NFCorpus Q | Represent the query from NutritionFacts to find a title with text of a medical document from PubMed about it |

| | |
|---|---|
| NFCorpus D | Represent this text of a medical document from PubMed to find a query someone may enter at NutritionFacts that it answers |
| NQ Q | Represent the Google search query to find an answer span from a Wikipedia article that addresses it |
| NQ D | Represent the Wikipedia article span to find a Google search query that would be addressed by it |
| SCIDOCS Q | Represent the scientific paper title to find the title with abstract of a scientific paper on PubMed that it has likely cited |
| SCIDOCS D | Represent the title with abstract of this scientific paper to find the title of another scientific paper on PubMed that likely cites this article |
| SciFact Q | Represent the scientific claim to find a scientific paper abstract from PubMed to support it |
| SciFact D | Represent the scientific paper abstract from PubMed to find a scientific claim that it supports |
| TRECCOVID Q | Represent the search query to find a scientific article about COVID-19 that adequately addresses the query |
| TRECCOVID D | Represent the scientific article about COVID-19 to find a user query that it adequately addresses |
| Touche2020 Q | Represent the question to find a title with passage of an argument from args.me that takes a stance about it |
| Touche2020 D | Represent the title with passage of an argument from args.me to find a question that it takes a stance about |
| QuoraRetrieval Q | Represent the Quora question to find another short duplicate question on Quora |
| QuoraRetrieval D | Represent the Quora question to find another short duplicate question on Quora |
| AskUbuntuDupQuestions Q | Represent the query to find a duplicate query on the AskUbuntu community forum |
| AskUbuntuDupQuestions D | Represent the query to find a duplicate query on the AskUbuntu community forum |
| MindSmallReranking Q | Represent the news headline to find another news headline that the same reader would enjoy |
| MindSmallReranking D | Represent the news headline to find another news headline that the same reader would enjoy |
| SciDocsRR Q | Represent the title to find a similar scientific paper title |
| SciDocsRR D | Represent the title to find a similar scientific paper title |
| StackOverflowDupQuestions Q | Represent the query to find a duplicate query on the StackOverflow Java/JavaScript/Python community forums |
| StackOverflowDupQuestions D | Represent the query to find a duplicate query on the StackOverflow Java/JavaScript/Python community forums |
| BIOSSES | Represent the text to find another biological statement with the same meaning |
| SICK-R | Represent the sentence to find another sentence with the same meaning |
| STS12 | Represent the sentence to find another sentence with the same meaning |

| | |
|---|---|
| STS13 | Represent the sentence to find another sentence with the same meaning |
| STS14 | Represent the sentence to find another sentence with the same meaning |
| STS15 | Represent the sentence to find another sentence with the same meaning |
| STS16 | Represent the sentence to find another sentence with the same meaning |
| STS17 | Represent the sentence to find another sentence with the same meaning |
| STS22 | Represent the sentence to find another sentence with the same meaning |
| STSBenchmark | Represent the sentence to find another sentence with the same meaning |
| SummEval Q | Represent the human-written summary to find a high-quality machine-written summary of the same news article |
| SummEval D | Represent the machine-written summary to find a human-written summary with similar quality of the same news article |

## U.2   EMBEDDING FEW-SHOT PROMPTS

Table 33: **1-shot example for the model trained on E5S.** The example is appended to the respective instruction in Table 31 separated by two newlines.

| Task Name | Instruction |
|---|---|
| Banking77Classification | For example given "I am still waiting on my card?", it would match with "card_arrival" |
| EmotionClassification | For example given "ive been feeling a little burdened lately wasnt sure why that was", it would match with "sadness" |
| ImdbClassification | For example given "If only to avoid making this type of film in the future. This film is interesting as an experiment but tells no cogent story.¡br /¿¡br /¿One might feel virtuous for sitting thru it because it touches on so many IMPORTANT issues but it does so without any discernable motive. The viewer comes away with no new perspectives (unless one comes up with one while one's mind wanders, as it will invariably do during this pointless film).¡br /¿¡br /¿One might better spend one's time staring out a window at a tree growing.¡br /¿¡br /¿", it would match with "negative" |
| BiorxivClusteringP2P | For example given "Association of CDH11 with ASD revealed by matched-gene co-expression analysis and mouse behavioral studies", it would match with "neuroscience" |
| TwitterSemEval2015 | For example given "The Ending to 8 Mile is my fav part of the whole movie", it would match with "Those last 3 battles in 8 Mile are THE shit" |

| | |
|---|---|
| TwitterURLCorpus | For example given "Liberals , dont let Donald Trump tarnish L.L. Beans sterling brand reputation ", it would match with "Liberals, Don’t Let Donald Trump Tarnish L.L. Bean’s Sterling Brand Reputation" |
| SprintDuplicateQuestions | For example given "Why is it impossible for me to find a easy way to send a picture with text on my Kyocera DuraCore ?", it would match with "Send or receive a picture with text - Kyocera DuraCore" |
| AskUbuntuDupQuestions | For example given "what is a short cut i can use to switch applications ?", you should retrieve "keyboard short cut for switching between two or more instances of the same application ?" |
| ArguAna | For example given "People will die if we don't do animal testing Every year, 23 new drugs are introduced in the UK alone.[13] Almost all will be tested on animals. A new drug will be used for a long time. Think of all the people saved by the use of penicillin. If drugs cost more to test, that means drug companies will develop less. This means more people suffering and dying", you should retrieve "animals science science general ban animal testing junior Many of these drugs are "me too" drugs – ones with a slight change that doesn't make much difference to an existing drug. [14] So often the benefits from animal testing are marginal, and even if there was a slight increase in human suffering, it would be worth it based on the animal suffering saved." |
| SCIDOCS | For example given "A Direct Search Method to solve Economic Dispatch Problem with Valve-Point Effect", you should retrieve "A Hybrid EP and SQP for Dynamic Economic Dispatch with Nonsmooth Fuel Cost Function Dynamic economic dispatch (DED) is one of the main functions of power generation operation and control. It determines the optimal settings of generator units with predicted load demand over a certain period of time. The objective is to operate an electric power system most economically while the system is operating within its security limits. This paper proposes a new hybrid methodology for solving DED. The proposed method is developed in such a way that a simple evolutionary programming (EP) is applied as a based level search, which can give a good direction to the optimal global region, and a local search sequential quadratic programming (SQP) is used as a fine tuning to determine the optimal solution at the final. Ten units test system with nonsmooth fuel cost function is used to illustrate the effectiveness of the proposed method compared with those obtained from EP and SQP alone." |
| STS12 | For example given "Counties with population declines will be Vermillion, Posey and Madison.", it would match with "Vermillion, Posey and Madison County populations will decline." |

| | |
|---|---|
| SummEval | The provided query could be "Mexican restaurant has decided to tap into $70 billion food delivery market. Fast-casual chain will work with the Postmates app to allow mobile orders. App works in similar way to Uber, using hired drivers to deliver the food. But the chain will add a 9% service charge - on top of Postmates\$5 rate." and the positive "chipotle has decided to tap into the $ 70 billion food delivery market by teaming up with an app to bring burritos straight to customers doors . the fast-casual chain will work with the postmates app to begin offering delivery for online and mobile orders in 67 cities . the restaurant plans to add a nine per cent service charge - with the delivery fees for postmates beginning at $ 5 and up depending on distance and demand ." |

Table 34: **1-shot example for the model trained on MEDI2.** The example is appended to the respective instruction in Table 32 separated by two newlines.

| Task Name | Instruction |
|---|---|
| Banking77Classification | The provided query could be "I am still waiting on my card?" and the positive "What can I do if my card still hasn't arrived after 2 weeks?" |
| EmotionClassification | The provided query could be "ive been feeling a little burdened lately wasnt sure why that was" and the positive "i feel like i have to make the suffering i m seeing mean something" |
| ImdbClassification | The provided query could be "If only to avoid making this type of film in the future. This film is interesting as an experiment but tells no cogent story.¡br /¿¡br /¿One might feel virtuous for sitting thru it because it touches on so many IMPORTANT issues but it does so without any discernable motive. The viewer comes away with no new perspectives (unless one comes up with one while one's mind wanders, as it will invariably do during this pointless film).¡br /¿¡br /¿One might better spend one's time staring out a window at a tree growing.¡br /¿¡br /¿" and the positive "The silent one-panel cartoon Henry comes to Fleischer Studios, billed as "The world's funniest human" in this dull little cartoon. Betty, long past her prime, thanks to the Production Code, is running a pet shop and leaves Henry in charge for far too long – five minutes. A bore." |
| SprintDuplicateQuestions | The provided query could be "Why is it impossible for me to find a easy way to send a picture with text on my Kyocera DuraCore ?" and the positive "Send or receive a picture with text - Kyocera DuraCore" |
| TwitterSemEval2015 | For example given "The Ending to 8 Mile is my fav part of the whole movie", it would match with "Those last 3 battles in 8 Mile are THE shit" |
| TwitterURLCorpus | For example given "Liberals , dont let Donald Trump tarnish L.L. Beans sterling brand reputation ", it would match with "Liberals, Don’t Let Donald Trump Tarnish L.L. Bean’s Sterling Brand Reputation" |

| | |
|---|---|
| AskUbuntuDupQuestions | The provided query could be "what is a short cut i can use to switch applications ?" and the positive "keyboard short cut for switching between two or more instances of the same application ?" |
| ArguAna | The provided query could be "People will die if we don't do animal testing Every year, 23 new drugs are introduced in the UK alone.[13] Almost all will be tested on animals. A new drug will be used for a long time. Think of all the people saved by the use of penicillin. If drugs cost more to test, that means drug companies will develop less. This means more people suffering and dying" and the positive "animals science science general ban animal testing junior Many of these drugs are "me too" drugs – ones with a slight change that doesn't make much difference to an existing drug. [14] So often the benefits from animal testing are marginal, and even if there was a slight increase in human suffering, it would be worth it based on the animal suffering saved." |
| SCIDOCS | The provided query could be "A Direct Search Method to solve Economic Dispatch Problem with Valve-Point Effect" and the positive "A Hybrid EP and SQP for Dynamic Economic Dispatch with Nonsmooth Fuel Cost Function Dynamic economic dispatch (DED) is one of the main functions of power generation operation and control. It determines the optimal settings of generator units with predicted load demand over a certain period of time. The objective is to operate an electric power system most economically while the system is operating within its security limits. This paper proposes a new hybrid methodology for solving DED. The proposed method is developed in such a way that a simple evolutionary programming (EP) is applied as a based level search, which can give a good direction to the optimal global region, and a local search sequential quadratic programming (SQP) is used as a fine tuning to determine the optimal solution at the final. Ten units test system with nonsmooth fuel cost function is used to illustrate the effectiveness of the proposed method compared with those obtained from EP and SQP alone." |
| STS12 | The provided query could be "Counties with population declines will be Vermillion, Posey and Madison." and the positive "Vermillion, Posey and Madison County populations will decline." |
| SummEval | The provided query could be "Mexican restaurant has decided to tap into $70 billion food delivery market. Fast-casual chain will work with the Postmates app to allow mobile orders. App works in similar way to Uber, using hired drivers to deliver the food. But the chain will add a 9% service charge - on top of Postmates$5 rate." and the positive "chipotle has decided to tap into the $ 70 billion food delivery market by teaming up with an app to bring burritos straight to customers doors . the fast-casual chain will work with the postmates app to begin offering delivery for online and mobile orders in 67 cities . the restaurant plans to add a nine per cent service charge - with the delivery fees for postmates beginning at $ 5 and up depending on distance and demand ." |

## U.3   GENERATIVE PROMPTS

Figure 14 until Figure 19 contain the prompts with examples used for our generative tasks.

---

**Input:**

---

```
<|user|>
```
The following are multiple choice questions (with answers) about abstract algebra.

Find the degree for the given field extension Q(sqrt(2), sqrt(3), sqrt(18)) over Q.
A. 0
B. 4
C. 2
D. 6
Answer:
```
<|assistant|>
```
The answer is:

---

**Correct completion:**

---

B

---

Figure 14: **MMLU prompt example.**

---

**Input:**

---

```
<|user|>
```
Answer the following questions.
Question: There are 15 trees in the grove. Grove workers will plant trees in the grove today. After they are done, there will be 21 trees. How many trees did the grove workers plant today?
Answer: There are 15 trees originally. Then there were 21 trees after some more were planted. So there must have been 21 - 15 = 6. So the answer is 6.

Question: If there are 3 cars in the parking lot and 2 more cars arrive, how many cars are in the parking lot?
Answer: There are originally 3 cars. 2 more cars arrive. 3 + 2 = 5. So the answer is 5.

Question: Leah had 32 chocolates and her sister had 42. If they ate 35, how many pieces do they have left in total?
Answer: Originally, Leah had 32 chocolates. Her sister had 42. So in total they had 32 + 42 = 74. After eating 35, they had 74 - 35 = 39. So the answer is 39.

Question: Jason had 20 lollipops. He gave Denny some lollipops. Now Jason has 12 lollipops. How many lollipops did Jason give to Denny?
Answer: Jason started with 20 lollipops. Then he had 12 after giving some to Denny. So he gave Denny 20 - 12 = 8. So the answer is 8.

Question: Shawn has five toys. For Christmas, he got two toys each from his mom and dad. How many toys does he have now?
Answer: Shawn started with 5 toys. If he got 2 toys each from his mom and dad, then that is 4 more toys. 5 + 4 = 9. So the answer is 9.

Question: There were nine computers in the server room. Five more computers were installed each day, from monday to thursday. How many computers are now in the server room?
Answer: There were originally 9 computers. For each of 4 days, 5 more computers were added. So 5 * 4 = 20 computers were added. 9 + 20 is 29. So the answer is 29.

Question: Michael had 58 golf balls. On tuesday, he lost 23 golf balls. On wednesday, he lost 2 more. How many golf balls did he have at the end of wednesday?
Answer: Michael started with 58 golf balls. After losing 23 on tuesday, he had 58 - 23 = 35. After losing 2 more, he had 35 - 2 = 33 golf balls. So the answer is 33.

Question: Olivia has $23. She bought five bagels for $3 each. How much money does she have left?
Answer: Olivia had 23 dollars. 5 bagels for 3 dollars each will be 5 x 3 = 15 dollars. So she has 23 - 15 dollars left. 23 - 15 is 8. So the answer is 8.

Question: The girls are trying to raise money for a carnival. Kim raises $320 more than Alexandra, who raises $430, and Maryam raises $400 more than Sarah, who raises $300. How much money, in dollars, did they all raise in total?
```
<|assistant|>
```
Answer:

---

**Correct completion:**

---

Kim raises 320+430=750 dollars. Maryam raises 400+300=700 dollars. They raise 750+430+400+700=2280 dollars. So the answer is 2280.

---

Figure 15: **GSM8K prompt example.**

**Input:**

```
<|user|>
```
Questions that involve enumerating objects and asking the model to count them.

Q: I have a blackberry, a clarinet, a nectarine, a plum, a strawberry, a banana, a flute, an orange, and a violin. How many fruits do I have?
A: Let's think step by step.
We first identify the fruits on the list and include their quantity in parentheses:
- blackberry (1)
- nectarine (1)
- plum (1)
- strawberry (1)
- banana (1)
- orange (1)
Now, let's add the numbers in parentheses: 1 + 1 + 1 + 1 + 1 + 1 = 6. So the answer is 6.

Q: I have an orange, a raspberry, two peaches, a blackberry, an apple, a grape, a nectarine, and three plums. How many fruits do I have?
A: Let's think step by step.
We first identify the fruits on the list and include their quantity in parentheses:
- orange (1)
- raspberry (1)
- peaches (2)
- blackberry (1)
- apple (1)
- grape (1)
- nectarine (1)
- plums (3)
Now, let's add the numbers in parentheses: 1 + 1 + 2 + 1 + 1 + 1 + 1 + 3 = 11. So the answer is 11.

Q: I have a lettuce head, a head of broccoli, an onion, a stalk of celery, two carrots, a garlic, and a yam. How many vegetables do I have?
A: Let's think step by step.
We first identify the vegetables on the list and include their quantity in parentheses:
- lettuce (1)
- broccoli (1)
- onion (1)
- celery (1)
- carrots (2)
- garlic (1)
- yam (1)
Now, let's add the numbers in parentheses: 1 + 1 + 1 + 1 + 2 + 1 + 1 = 8. So the answer is 8.

Q: I have a banana, four strawberries, an apple, two peaches, a plum, a blackberry, and two raspberries. How many fruits do I have?
```
<|assistant|>
```

**Correct completion:**

12

Figure 16: **BBH prompt example.**

---

**Input:**

---

```
<|user|>
```
Jawab pertanyaan berikut berdasarkan informasi di bagian yang diberikan.

Bagian: Mula-mula pada pelukis seorang pelukis pemandangan Wahdi Sumanta, Abdullah Suriosubroto (ayah Basuki Abdullah). Kemudian bertemu dan berkenalan dengan Affandi, Sudarso, dan Barli. Mereka lalu membentuk kelompok Lima serangkai. Di rumah tempat tinggal Affandi mereka mengadakan latihan melukis bersama dengan tekun dan mendalam. Dari Wahdi, ia banyak menggali pengetahuan tentang melukis. Kegiatannya bukan hanya melukis semata, tetapi pada waktu senggang ia menceburkan diri pada kelompok sandiwara Sunda sebagai pelukis dekor. Dari pengalaman itulah, ia mengasah kemampuannya.
Pertanyaan: dari manakah Hendra Gunawan belajar melukis?
Jawaban: kelompok Lima serangkai

Bagian: Empat Sehat Lima Sempurna adalah kampanye yang dilakukan pemerintah sejak tahun 1955 untuk membuat masyarakat memahami pola makan yang benar.[1]. Dalam konsep 4 sehat 5 sempurna, makanan dibagi atas empat sumber nutrisi penting, yaitu makanan pokok, lauk pauk, sayur-mayur, buah-buahan, dan disempurnakan dengan susu bila mampu, menjadi lima sempurna[2] Konsep ini menekankan pentingnya empat golongan makanan berupa sumber kalori untuk tenaga, protein untuk pembangun, sayur dan buah sumber vitamin dan mineral untuk pemeliharaan.[1]
Pertanyaan: siapakah yang mencptakan Ide 4 sehat 5 sempurna pertama kali?

```
<|assistant|>
```
Jawaban:

---

**Correct completion:**

---

pemerintah

---

Figure 17: **TyDi QA prompt example from Indonesian.**

**Input:**

```
<|user|>
```
Write a Python function 'has_close_elements(numbers: List[float], threshold: float) -¿ bool' to solve the following problem: Check if in given list of numbers, are any two numbers closer to each other than given threshold.

```
>>> has_close_elements([1.0, 2.0, 3.0], 0.5)
False
>>> has_close_elements([1.0, 2.8, 3.0, 4.0, 5.0, 2.0], 0.3)
True
```

```
<|assistant|>
```

```python
from typing import List
def has_close_elements(numbers: List[float], threshold: float) -> bool:
    """ Check if in given list of numbers, are any two numbers closer to
        each other than
    given threshold.
    >>> has_close_elements([1.0, 2.0, 3.0], 0.5)
    False
    >>> has_close_elements([1.0, 2.8, 3.0, 4.0, 5.0, 2.0], 0.3)
    True
    """
```

**Correct completion:**

```python
    for idx, elem in enumerate(numbers):
        for idx2, elem2 in enumerate(numbers):
            if idx != idx2:
                distance = abs(elem - elem2)
                if distance < threshold:
                    return True

    return False
```

Figure 18: **HumanEvalSynthesize prompt example.**

**Input:**

```
<|user|>
```
If you were a Shakespearean character, how would you declare your love for someone in a soliloquy?

**Completion by GPT-4 Turbo:**

As an AI, I lack personal emotions and experiences, but I can craft a soliloquy in the style of a Shakespearean character declaring love. Here's an example:

(Imagine a dimly lit stage, a single spotlight shining down upon the forlorn lover, who steps forward to share the depths of their heart.)

O, speak, thou lustrous beacon of the night,
With stars enkindled in thine eyes so bright.
Thou art the daybreak to my endless gloom,
The rose that in my barren heart doth bloom.

Forsooth, this love, a tempest full of grace,
Hath swept upon my soul without a trace.
My heart, once caged, now beats beneath thy gaze,
And yearns to laud thy beauty all its days.

What art thou, Love? A tyrant or a muse?
That rends my sense and all my thoughts confuse?
Thou art the riddle none can e'er resolve,
Yet in thy puzzle, my life doth revolve.

With every breath, I thee do magnify,
And to the world, my passion testify.
Thine essence hath become my sole refrain,
A melody that in my mind doth reign.

Shall I compare thee to a summer's day?
Thou art more lovely and more temperate.
Rough winds do shake the darling buds of May,
And summer's lease hath all too short a date.

But thy eternal summer shall not fade,
Nor lose possession of that fair thou ow'st.
Nor shall Death brag thou wander'st in his shade,
When in eternal lines to time thou grow'st.

So long as men can breathe, or eyes can see,
So long lives this, and this gives life to thee.

(With a heavy sigh, the lover steps back, their heart laid bare, the words hanging in the air like the fragrance of an unseen bloom, their love declared, though the object of their affection remains unaware, the audience enthralled by the vulnerability of true, unspoken love.)

Figure 19: **AlpacaEval prompt example.**

U.4   RAG PROMPTS

**Generative input:**

```
<|user|>
```
what was the name of darth vader star destroyer
```
<|assistant|>
```
The answer is

**Generation:**

 not clear. Darth Vader had a star destroyer, but it was

Figure 20: **No RAG example and generation by GRITLM 7B.**

**Generative input:**

```
<|user|>
```
what was the name of darth vader star destroyer

Star Destroyer The iconic Star Destroyer first appears in the opening scene of Star Wars, as Darth Vader's flagship, the Devastator, chases the Tantive IV above Tatooine. This appearance shows the Imperial ship's massive size in comparison to the Tantive IV.

Optionally using the prior context answer the query prior to it
```
<|assistant|>
```
The answer is

**Generation:**

 the Devastator.``

Figure 21: **RAG query then document example and generation by GRITLM 7B.**

**Generative input:**

```
<|user|>
```
Star Destroyer The iconic Star Destroyer first appears in the opening scene of Star Wars, as Darth Vader's flagship, the Devastator, chases the Tantive IV above Tatooine. This appearance shows the Imperial ship's massive size in comparison to the Tantive IV.

what was the name of darth vader star destroyer

Answer the prior query while optionally using the context prior to it
```
<|assistant|>
```
The answer is

**Generation:**

the Devastator.``

Figure 22: **RAG document then query example and generation by GRITLM 7B.**

**Embedding input (query) passed via key-value states:**

```
<|embed|>
```
what was the name of darth vader star destroyer

**Generative input:**

```
\n<|user|>
```
Star Destroyer The iconic Star Destroyer first appears in the opening scene of Star Wars, as Darth Vader's flagship, the Devastator, chases the Tantive IV above Tatooine. This appearance shows the Imperial ship's massive size in comparison to the Tantive IV.

Optionally using the prior context answer the query prior to it
```
<|assistant|>
```
The answer is

**Generation:**

Star Destroyer.``

Figure 23: **GRIT Query Caching example and generation by GRITLM 7B.**

**Embedding input (doc) passed via key-value states and cached in the index:**

```
<|embed|>
```
Star Destroyer The iconic Star Destroyer first appears in the opening scene of Star Wars, as Darth Vader's flagship, the Devastator, chases the Tantive IV above Tatooine. This appearance shows the Imperial ship's massive size in comparison to the Tantive IV.

**Generative input:**

```
\n<|user|>
```
what was the name of darth vader star destroyer

Answer the prior query while optionally using the context prior to it
```
<|assistant|>
```
The answer is

**Generation:**

Devastator. The iconic Star Destroyer first appears in the opening

Figure 24: **GRIT Doc Caching example and generation by GRITLM 7B.**

**Embedding input (doc) passed via key-value states and cached in the index:**

```
<|embed|>
```
Star Destroyer The iconic Star Destroyer first appears in the opening scene of Star Wars, as Darth Vader's flagship, the Devastator, chases the Tantive IV above Tatooine. This appearance shows the Imperial ship's massive size in comparison to the Tantive IV.

**Embedding input (query) passed via key-value states:**

```
<|embed|>
```
what was the name of darth vader star destroyer

**Generative input:**

```
\n<|user|>
```
Answer the prior query while optionally using the context prior to it
```
<|assistant|>
```
The answer is

**Generation:**

the Star Destroyer. The Star Destroyer is a massive spacecraft

Figure 25: **GRIT Doc-Query Caching example and generation by GRITLM 7B.** Unlike for Doc Caching, we prepend the bos token ("$< s >$") to both query and document, which improved the match score from 14.13 to 18.39.

---

**Embedding input (query) passed via key-value states:**

---

```
<|embed|>
```
what was the name of darth vader star destroyer

---

**Embedding input (doc) passed via key-value states and cached in the index:**

---

```
<|embed|>
```
Star Destroyer The iconic Star Destroyer first appears in the opening scene of Star Wars, as Darth Vader's flagship, the Devastator, chases the Tantive IV above Tatooine. This appearance shows the Imperial ship's massive size in comparison to the Tantive IV.

---

**Generative Input:**

---

```
\n<|user|>
```
Optionally using the prior context answer the query prior to it
```
<|assistant|>
```
The answer is

---

**Generation:**

---

the Star Destroyer.

---

Figure 26: **GRIT Query-Doc Caching example and generation by GRITLM 7B.**

## V  HARDWARE

For the training of GRITLM 7B, we used 8 nodes with 8 NVIDIA A100 80GB GPUs each for 48 hours corresponding to 3,072 GPU hours. Meanwhile for GRITLM 8x7B, we used 32 nodes with 8 NVIDIA H100 80GB GPUs each for 80 hours corresponding to 20,480 GPU hours. As we train both models for 1253 steps, this corresponds to several minutes per step. This slow training time is mainly due to (a) a large batch size per step, (b) large models and our associated strategies to make them fit into memory at the cost of speed (Appendix L, Appendix M), and (c) a cluster with slow inter-node communication. The Gen.-only and Emb.-only models in Table 1 used 72 and 1760 H100 80GB GPU hours, respectively. Adding up all ablations and evaluations, we likely used somewhere around 100,000 GPU hours.

## W  ARTIFACTS

Table 35: **Produced artifacts.**

| Artifact | Public Link |
|---|---|
| *Table 6* | |
| 7B KTO | https://hf.co/GritLM/GritLM-7B-KTO |
| 8x7B KTO | https://hf.co/GritLM/GritLM-8x7B-KTO |
| *Table 10* | |
| CCCC WM | https://hf.co/GritLM/gritlm_m7_sq2048_medi |
| CCCC LT | https://hf.co/GritLM/gritlm_m7_sq2048_medi_lasttoken |

| BBCC M | `https://hf.co/GritLM/gritlm_m7_sq2048_medi_bbcc` |
|---|---|

*Table 11*

| CC WM | `https://hf.co/GritLM/emb_m7_sq2048_medi` |
|---|---|
| CB M | `https://hf.co/GritLM/emb_m7_sq2048_medi_cb` |
| BB M | `https://hf.co/GritLM/emb_m7_sq2048_medi_bb` |

*Table 12*

| CC | `https://hf.co/GritLM/gen_m7_sq2048_tulu2_ep1` |
|---|---|
| BC | `https://hf.co/GritLM/gen_m7_sq2048_tulu2_bc` |
| BC IL | `https://hf.co/GritLM/gen_m7_sq2048_tulu2_bcil` |

*Table 13*

| Mistral 7B | `https://hf.co/GritLM/gritlm_m7_sq1024_st100_zephfmt` |
|---|---|
| Llama 2 7B | `https://hf.co/GritLM/gritlm_l7_sq1024_st100_zephfmt` |
| GPT-J 6B | `https://hf.co/GritLM/gritlm_g6_sq1024_st100_zephfmt` |

*Table 14*

| MEDI | `https://hf.co/GritLM/emb_m7_sq2048_medi` |
|---|---|
| MEDI2 NNI | `https://hf.co/GritLM/emb_m7_sq2048_medi2nni` |
| MEDI2 | `https://hf.co/GritLM/emb_m7_sq2048_medi2` |
| MEDI2 + W | `https://hf.co/GritLM/emb_m7_sq2048_medi2weights` |

*Table 15*

| MEDI | `https://hf.co/GritLM/gritlm_m7_sq2048_medi` |
|---|---|
| MEDI2 | `https://hf.co/GritLM/gritlm_m7_sq2048_medi2` |
| BBCC MEDI | `https://hf.co/GritLM/gritlm_m7_sq2048_medi_bbcc` |
| BBCC MEDI2 | `https://hf.co/GritLM/gritlm_m7_sq2048_medi2_bbcc` |
| BBCC MEDI2BGE | `https://hf.co/GritLM/gritlm_m7_sq2048_medi2bge_bbcc` |
| BBCC E5 | `https://hf.co/GritLM/gritlm_m7_sq2048_e5_bbcc` |

*Table 16*

| Tülu 2 1 EP | `https://hf.co/GritLM/gen_m7_sq2048_tulu2_ep1` |
|---|---|
| Tülu 2 2 EP | `https://hf.co/GritLM/gen_m7_sq2048_tulu2_ep2` |
| OASST 1 EP | `https://hf.co/GritLM/gen_m7_sq2048_oasst_ep1` |
| OASST 2 EP | `https://hf.co/GritLM/gen_m7_sq2048_oasst_ep2` |
| UltraChat | `https://hf.co/GritLM/gen_m7_sq2048_ultrachat_ep1` |

*Table 17*

| No head | `https://hf.co/GritLM/gritlm_m7_sq2048_medi_gendups` |
|---|---|

| -> 1024 | `https://hf.co/GritLM/gritlm_m7_sq2048_medi_proj1024_gendups` |
|---|---|

| *Table 18* | |
|---|---|
| 256 | `https://hf.co/GritLM/gritlm_m7_sq2048_medi2` |
| 4096 | `https://hf.co/GritLM/gritlm_m7_sq2048_medi2_bs4096` |

| *Table 19* | |
|---|---|
| BF16 | `https://hf.co/GritLM/gritlm_m7_sq2048_e5s_bbcc_bs2048_token6` |
| FP32 | `https://hf.co/GritLM/gritlm_m7_sq2048_e5s_bbcc_bs2048_token6_fp32` |

| *Table 20* | |
|---|---|
| Any dataset | `https://hf.co/GritLM/gritlm_m7_sq2048_e5_bbcc_token_anyneg` |
| Same dataset | `https://hf.co/GritLM/gritlm_m7_sq2048_e5_bbcc_token` |

| *Table 21* | |
|---|---|
| Tülu 2 | `https://hf.co/GritLM/gen_m7_sq2048_tulu2_ep1` |
| Zephyr $\beta$ | `https://hf.co/GritLM/gen_m7_sq2048_tulu2_ep1_zephfmt` |

| *Table 22* | |
|---|---|
| MEDI 2048 | `https://hf.co/GritLM/gritlm_m7_sq2048_medi` |
| MEDI 4096 | `https://hf.co/GritLM/gritlm_m7_sq4096_medi` |
| BBCC MEDI2 512 | `https://hf.co/GritLM/gritlm_m7_sq512_medi2_bbcc` |
| BBCC MEDI2 2048 | `https://hf.co/GritLM/gritlm_m7_sq2048_medi2_bbcc` |

| *Table 23* | |
|---|---|
| E5 Token 4.2 | `https://hf.co/GritLM/gritlm_m7_sq2048_e5s_bbcc_bs2048_token4` |
| E5 Token 6.0 | `https://hf.co/GritLM/gritlm_m7_sq2048_e5s_bbcc_bs2048_token6` |
| E5 Mix 32 -> 8 | `https://hf.co/GritLM/GritLM-7b` |
| MEDI2 Mix 4 -> 64 | `https://hf.co/GritLM/gritlm_m7_sq2048_medi2_bbcc` |
| MEDI2 Mix 32 -> 8 | `https://hf.co/GritLM/gritlm_m7_sq2048_medi2_bbcc_bs4096` |

| *Other* | |
|---|---|
| Code | `https://github.com/ContextualAI/gritlm` |
| Logs | `https://wandb.ai/muennighoff/gritlm` |
| Tülu 2 | `https://hf.co/datasets/GritLM/tulu2` |
| MEDI | `https://hf.co/datasets/GritLM/medi` |

| | |
|---|---|
| MEDI2 | `https://hf.co/datasets/GritLM/medi2` |
| MEDI2BGE | `https://hf.co/datasets/GritLM/medi2bge` |
| GRITLM 7B NQ Index (§5) | `https://hf.co/datasets/GritLM/index` |

| *Main artifacts* | |
|---|---|
| GRITLM 7B | `https://hf.co/GritLM/GritLM-7B` |
| GRITLM 8x7B | `https://hf.co/GritLM/GritLM-8x7B` |

Table 36: **Used artifacts released by others.**

| Model / Dataset | Public Link |
|---|---|
| GPT-4 (OpenAI et al., 2023) | `https://openai.com/gpt-4` |
| OpenAI v3 (OpenAI et al., 2023) | `https://openai.com/blog/new-embedding-models-and-api-updates` |
| Gemini (Team et al., 2023) | `https://deepmind.google/technologies/gemini/` |
| Llama 2 (Touvron et al., 2023) | `https://hf.co/meta-llama` |
| Mistral 7B (Jiang et al., 2023) | `https://hf.co/mistralai/Mistral-7B-v0.1` |
| Mistral 7B Instruct (Jiang et al., 2023) | `https://hf.co/mistralai/Mistral-7B-Instruct-v0.1` |
| Mixtral 8x7B (Jiang et al., 2024) | `https://hf.co/mistralai/Mixtral-8x7B-v0.1` |
| Mixtral 8x7B Instruct (Jiang et al., 2024) | `https://hf.co/mistralai/Mixtral-8x7B-Instruct-v0.1` |
| Tülu 2 (Ivison et al., 2023) | `https://hf.co/collections/allenai/tulu-v2-suite-6551b56e743e6349aab45101` |
| GPT-J 6B (Wang & Komatsuzaki, 2021) | `https://hf.co/EleutherAI/gpt-j-6b` |
| SGPT BE 5.8B (Muennighoff, 2022) | `https://hf.co/Muennighoff/SGPT-5.8B-weightedmean-msmarco-specb-bitfit` |
| Instructor-XL 1.5B (Su et al., 2023) | `https://hf.co/hkunlp/instructor-xl` |
| BGE Large 0.34B (Xiao et al., 2023) | `https://hf.co/BAAI/bge-large-en-v1.5` |
| Zephyr 7B $\beta$ (Tunstall et al., 2023) | `https://hf.co/HuggingFaceH4/zephyr-7b-beta` |
| E5 Mistral 7B (Wang et al., 2024) | `https://hf.co/intfloat/e5-mistral-7b-instruct` |
| UltraChat (Ding et al., 2023; Tunstall et al., 2023) | `https://hf.co/datasets/HuggingFaceH4/ultrachat_200k` |
| OASST (Köpf et al., 2023; Muennighoff et al., 2023a) | `https://hf.co/datasets/bigcode/oasst-octopack` |

