# OpenReview forum: "Generative Representational Instruction Tuning"
_ICLR.cc/2025/Conference — ICLR 2025 Poster_

### Official Review · Reviewer_xDdr · 2024-11-03

**Soundness:** 4
**Presentation:** 4
**Contribution:** 4
**Rating:** 8
**Confidence:** 3

**Summary:**

The paper introduces a new framework called GRIT, which aims to unify text generation and embedding tasks within a single LLM, GRITLM. The model handles both tasks efficiently by distinguishing between them through instructions, which streamline use in multi-task applications like RAG. The authors demonstrate that GRITLM performs strongly on text representation and generation benchmarks, achieving competitive performance on the Massive Text Embedding Benchmark (MTEB) while also excelling in generative tasks.

Contributions:
1. Unified Generative and Embedding Model: The GRIT framework combines generative and embedding tasks within a single LLM. By using instructional prompts to distinguish between tasks, GRIT allows both generation and embedding without sacrificing performance. GRIT also reduces the need for separate models and complex infrastructure setups. This unification could simplify real-world deployments, particularly for applications that traditionally require both retrieval and generation components, such as search engines, recommendation systems, and conversational AI.
2. Efficient RAG catching design: The paper proposes innovative caching techniques, like Doc-Query Caching, and Query-Doc Caching, that significantly speed up RAG processes by reducing the number of forward passes required for long document processing. This approach reduces computational load for RAG tasks, enhancing efficiency in applications that rely on fast, context-sensitive retrieval and generation.
3. Competitive Performance Across Generative and Embedding Benchmarks: GRITLM achieves strong results on both the MTEB and several generative tasks, outperforming other open models of comparable size. This dual-task proficiency demonstrates that GRIT can match or exceed task-specific models, marking a significant step toward a general-purpose language model that handles both types of tasks seamlessly.
4. Task-Specific Performance Optimization: GRIT introduces several improvements, such as bidirectional attention with mean pooling for embedding tasks and mixed token-sample level loss aggregation for generative tasks. These innovations contribute to the model's performance across diverse tasks and offer insights into optimizing large language models for multi-task functionality.

**Strengths:**

Originality: The GRIT framework presents an original approach by unifying generative and representational capabilities within a single model, GRITLM, that can seamlessly switch between tasks based on instructional prompts. This concept is innovative as it directly addresses a long-standing limitation in language models: the need for distinct models optimized separately for generation and embedding. Previous work has focused on either generation or embedding, often leading to complex infrastructures where multiple models must be managed, synchronized, and deployed separately. GRIT’s unified approach not only simplifies these workflows but also brings both task types under one architecture without compromising performance. Additionally, GRIT’s application of caching techniques to accelerate RAG showcases an innovative use of model design to enhance efficiency, a departure from traditional RAG approaches that rely on separate models.

Quality: The paper demonstrates a strong methodological foundation, supported by comprehensive experimentation and ablation studies. The authors provide detailed evaluations on major benchmarks, contrasting GRITLM’s performance with task-specific models to validate its efficacy as a multi-task solution. The robustness of the results is further confirmed through comparisons with proprietary models and current open-source alternatives, evidencing GRIT’s strong performance in both generative and embedding tasks. The use of ablations to explore trade-offs in task prioritization, loss aggregation, and memory efficiency contributes to the overall rigor, allowing readers to clearly understand how GRIT’s dual-objective structure was optimized. The paper’s experiments on efficiency gains with caching also underscore the quality of its findings, providing quantitative backing for its claims regarding speed improvements in RAG tasks.

Clarity: The paper is well-organized and clear in its presentation, guiding the reader through complex ideas with a logical flow. Key concepts, GRIT’s caching mechanisms, and instructional tuning, are introduced with adequate background and broken down into understandable segments. Figures effectively support comprehension, making the technical details more accessible. The thorough presentation of results, including detailed tables and ablation analyses, provides clarity around GRIT’s performance relative to baselines, demonstrating where it excels and where there may be trade-offs. Additionally, the inclusion of an in-depth Appendix suggests a commitment to transparency and accessibility, ensuring that interested readers have the resources to delve deeper into implementation specifics and experiment configurations.

Significance: The significance of GRIT lies in its potential to impact the field of NLP by simplifying multi-task language model deployment and reducing reliance on separate models for embedding and generation tasks. Furthermore, GRIT’s caching innovations for RAG tasks significantly reduce computational overhead and latency, especially in long-document settings, which is valuable for any application that relies on fast, context-aware responses. Moreover, GRIT’s design choices and improvements, such as mean pooling for embeddings and loss aggregation, may inspire further research into architectural unification across other language model tasks.

**Weaknesses:**

Storage Costs for Caching: The paper proposes innovative caching strategies to speed up RAG, but for example, Doc Caching, in particular, requires 30TB of storage for key-value states on GRITLM 7B. Such high storage demands are prohibitive in many real-world scenarios, limiting the practical usefulness of these techniques.

Instruction Dependence: The model’s reliance on instruction-based differentiation between tasks could lead to inconsistent performance if instructions are poorly structured or if the model misinterprets the intended task. Instruction-based models can sometimes be sensitive to variations in phrasing, and such dependency on clear, well-defined instructions might limit GRIT’s robustness in noisy or ambiguous real-world applications.

Complexity of Caching Mechanisms and Trade-offs: While the caching mechanisms offer significant speed-ups, they introduce substantial complexity to the model's architecture and inference workflow. The paper acknowledges that Query-Doc Caching can result in degraded performance due to mismatches in attention patterns. This complexity could make it challenging for practitioners to implement GRITLM optimally and may lead to inconsistent performance across different tasks and input types.

**Questions:**

Storage Costs for Caching: The caching techniques provide impressive speed improvements, but the storage requirements (e.g., 30TB for Doc Caching) are substantial. Can the authors discuss potential strategies to make these techniques more feasible for real-world use, particularly in terms of storage optimization?

Instruction Dependence: Including experimental results on GRITLM’s sensitivity to instruction phrasing and format would provide valuable insights into its robustness and areas for improvement.

---

> ### Author Response · Authors · 2024-11-19
>
> Thanks a lot for your extensive review and highlighting the originality and novelty of the approach.
>
> **Storage Costs:** Query caching does not require any additional storage, only doc caching requires storing additional key-value states. For doc caching the KV cache can be fully offloaded to disk and does not need to be kept in memory. Disk storage is generally cheap. One can also store only part of the cache, e.g. only cache the key-value states of the first N layers. This will still lead to speed-ups but to a lesser extent. Thus, practitioners can flexibly choose the amount of storage they want to use depending on their specific setup.
>
> **Instruction Dependence:** GritLM can produce reliable embeddings without instructions – in fact, for embedding documents during retrieval we do not use instructions. The generative part, however, indeed needs instructions. There has been prior work investigating the robustness of generative instruction-tuned models [1] and while it can be problematic at small scale, these issues generally go away at larger scales.
>
> **Complexity of Caching:** Note that caching does not change the architecture of the model. It does add some additional steps at inference, however, these are only ~3 lines of additional code to extract the key-value cache and repass it to the model. We provide open-source example code for running the caching in the supplementary material. Overall, we think that the caching is less complex than having to load and serve a second embedding/generative model as is necessary for current RAG setups that do not use GRIT.
>
> [1] Evaluating the zero-shot robustness of instruction-tuned language models by J Sun, C Shaib, BC Wallace, URL https://arxiv.org/abs/2306.11270

---

> > ### Author Response · Authors · 2024-11-25
> > **Follow-up**
> >
> > Dear Reviewer,
> >
> > We'd appreciate it if you'd let us know if our response has addressed your concerns.
> >
> > Thank you!

---

> > > ### Comment · Reviewer_xDdr · 2024-11-29
> > > **Response to the authors**
> > >
> > > Thank you for following up! Yes, your response has addressed my concerns. I appreciate the detailed clarifications provided.

---

### Official Review · Reviewer_2rfu · 2024-11-04

**Soundness:** 3
**Presentation:** 3
**Contribution:** 3
**Rating:** 6
**Confidence:** 3

**Summary:**

This work introduces GRIT, a method to train a single large language model to excel at both generative and embedding tasks through differentiated instructions. They proposed GRITLM models achieve SOTA performance on the Massive Text Embedding Benchmark and surpass other models in generative tasks. GRIT unifies the two tasks without compromising performance, offering efficiency gains such as over 60% faster Retrieval-Augmented Generation for long documents. The unified model simplifies infrastructure by handling both embedding and generative tasks, reducing the need for separate models.

**Strengths:**

1. GRIT introduces a novel approach that enables a single large language model to excel at both generative and embedding tasks, traditionally handled separately.
2. By eliminating the need for separate retrieval and generation models, GRIT speeds up RAG by more than 60% for long documents, which is a substantial improvement in processing time and resource management

**Weaknesses:**

1. The unified model requires more training resource, while there is no comparison of the performance between separate generation models and embedding models under the same resource consumption as shown in Table 1 and Table 2.
2. The paper uses the Mistral model as the base model. I think it would also be necessary to conduct experiments on the LLaMA series models to verify the robustness of the method.

**Questions:**

Please refer to the above.

---

> ### Author Response · Authors · 2024-11-19
>
> Thanks a lot for your review and notes on GritLMs strong performance.
>
> **Comparison of training resources:** Great point! As we write in Lines 114-118, we believe that finetuning is so cheap compared to pretraining, that the additional training resources for GRIT don’t make a big difference. However, it may still matter in resource-constrained scenarios, thus we have added precise information on the GPU hours for each approach. Specifically: We used 72 GPU hours for the gen-only model, 1760 GPU hours for the emb-only model and 3072 GPU hours for GRIT. The GRIT number was already in “Appendix P: Hardware” and we have added the other two numbers there, too. Increasing the GPU hours for the gen-/emb-only model to match GRIT is unlikely to improve performance as all models have converged; Especially for the gen-only model, it would probably just lead to an excessive number of epochs. Nonetheless, we acknowledge that efficiency is a limitation and we have added more discussion on this in our “Appendix Q: Limitations and Future Work”, where we mention that packing and reusing the same samples for both the embedding and generative losses could significantly improve efficiency. We have uploaded the revised paper with the resource numbers and additional discussion - thank you for bringing this up!
>
> **Other base models:** We experimented with different base models (Llama2 and GPT-J) in Appendix A, Table 5, where we found that the approach works just as well but Mistral delivers better performance. In addition, we have also finetuned Mixtral using GRIT. All of these variants will be open-sourced.

---

> > ### Author Response · Authors · 2024-11-25
> > **Follow-up**
> >
> > Dear Reviewer,
> >
> > We'd appreciate it if you'd let us know if our response has addressed your concerns.
> >
> > Thank you!

---

### Official Review · Reviewer_vUnt · 2024-11-04

**Soundness:** 3
**Presentation:** 3
**Contribution:** 2
**Rating:** 6
**Confidence:** 3

**Summary:**

This paper introduces GritLM, a language model designed to excel at both text generation and embedding tasks.  Current large language models (LLMs) typically specialize in one or the other, requiring separate models for applications that need both functionalities. GritLM addresses this limitation by employing a joint training approach.

The model architecture leverages a standard autoregressive generation head for text generation, trained with a next-token prediction cross-entropy loss.  For embedding tasks, GritLM uses bidirectional encoding of the input prompt and mean pooling of the final hidden layer representations.  A contrastive loss with in-batch negatives is applied to these embedding representations. The overall training objective combines these two losses, allowing the model to learn both tasks concurrently.

Experimental results demonstrate that GritLM achieves competitive performance on both generation and embedding benchmarks, comparable to similarly-sized specialized models.  Furthermore, the authors explore the benefits of this unified architecture in two specific scenarios: (1) reranking, where GritLM improves its own generated text through its embedding capabilities, and (2) retrieval-augmented generation (RAG), where the unified model serves as both retriever and reader, significantly reducing inference costs.

**Strengths:**

* GritLM effectively demonstrates strong performance in both generation and embedding tasks within a single model.

* The paper presents a thorough experimental evaluation, including reranking and RAG scenarios, showcasing the practical advantages of the unified architecture.

**Weaknesses:**

The scalability of the proposed method raises some concerns. The practicality of training and deploying a single model for both retrieval and generation may be limited to certain model sizes. In real-world applications, employing a smaller, faster embedding model alongside a potentially much larger generation model is often preferred. A smaller embedding model typically suffices for retrieval, while larger generation models are crucial for high-quality text generation. The paper would benefit from a discussion addressing the impact of model scale on the effectiveness of the unified approach and whether it remains advantageous when using vastly different-sized models for retrieval and generation.  Specifically, quantifying the trade-off between performance and efficiency in such mixed-size scenarios would strengthen the paper's claims.

(An alternative approach for using different sizes of embedder and generator is to use the output of the N-th layer (where N is relatively small) for embeddings instead of the last layer.)

**Questions:**

N/A

---

> ### Author Response · Authors · 2024-11-19
>
> Thank you for your detailed review and highlighting the extensive experiments!
>
> **Scalability:** GritLM-7B is faster than a 7B generative model + a tiny embedding model for RAG when using the caching techniques we introduce. This is because the caching techniques (e.g. doc caching) will only require a single forward pass of GritLM-7B at inference, while in the other case, a forward pass for both the 7B model and the tiny embedding model is required. Without the caching techniques, speed indeed matters. We like your idea of using an intermediate layer and would expect it to lead to a performance drop while improving speed. In fact, we performed a similar experiment to reduce storage costs in Appendix A, Table 5 (e), where we find that we can downproject the embeddings to a 4x smaller dimension (->1024) at a small reduction in performance. Similarly, if we cannot use caching, we could increase speed 2x by taking the output of the middle layer at a slight reduction in performance. We have added a short note on this in Appendix A, thanks a lot for bringing this to our attention!

---

> > ### Author Response · Authors · 2024-11-25
> > **Follow-up**
> >
> > Dear Reviewer,
> >
> > We'd appreciate it if you'd let us know if our response has addressed your concerns.
> >
> > Thank you!

---

### Official Review · Reviewer_XQzj · 2024-11-06

**Soundness:** 3
**Presentation:** 4
**Contribution:** 3
**Rating:** 8
**Confidence:** 4

**Summary:**

The paper presents generative representational instruction tuning (GRIT), a unified model for embedding and generative tasks in text. GRIT learns embedding representations with a bidirectional attention followed by mean pooling and a instruction tuning with causal attention. The experiments show that GRITLM outperforms various prior open models on the MTEB benchmark and matches the performance of several instruction tuning models. Furthermore, the unified model speeds up retrieval augmented generation by 60%.

**Strengths:**

- The paper is very well written. It is easy to follow the main motivation of the paper. The related work positions the paper well.
- The paper presents large-scale experiments. The model matches the performance of strong baselines on challenging benchmarks such as MTEB and instruction tuning datasets.
- The caching mechanism reduces latency for RAG, especially longer sequences.
- The GRITLM model will be useful for practitioners.

**Weaknesses:**

**Mixed results**
The main contribution of the unified method is to reduce the latency for generating the output. However, Table 4 shows a tradeoff between performance and latency. In Doc-Query and Query-Doc experiments, we see that GRITLM speeds up RAG but at the cost of overall performance. Furthermore,  GRITLM does not show significant speed-ups on GPUs. Finally, I would be curious to see if a smaller embedding model (besides a smaller GRITLM) shows improved performance compared to the RAG performance in Table 4.

**Modularity.**
One of the main advantages of RAG is that it is modular. The separation of the embedding model and the generative model makes it easy to swap out either one of the components. With a unified embedding and generative model, the entire model has to be retrained which can be computationally expensive.

**Include more recent work**
The authors have acknowledged that the more recent embedding models, such as NV-Embed, show improved performance over GRITLM. It would be awesome if the authors cited more recent work [a, b] and more.

[a] ​​SFR-Embedding-Mistral:Enhance Text Retrieval with Transfer Learning.

[b] Towards General Text Embeddings with Multi-stage Contrastive Learning

**Questions:**

Please see the weaknesses.

---

> ### Author Response · Authors · 2024-11-19
>
> Thank you for your detailed review. We are glad that you think the model will be useful and the work is well-positioned!
>
> **Mixed results:** As highlighted in the text we generally recommend doc caching (or query caching) but not the combined doc-query / query-doc caching mechanisms. We mostly present the query-doc and doc-query variants to inspire future work and are working on improving their performance in follow-up work. In Figure 5, we show that caching reduces latency on GPUs by around half compared to traditional RAG, which can be quite significant in time-sensitive applications. We note that the speed-up from doc (query) caching correlates directly with the length of documents (queries). E.g. for a book retrieval service where books are retrieved given user queries and each book has on the order of 10,000 or more tokens, the speed-up via doc caching would be significantly more than 2x, probably closer to 10x (depending on the query lengths). We ran RAG with a smaller model in Table 4, specifically, we ran using BGE as the embedding model which we also compare retrieval performance with in Table 1. The generative model is still GritLM-7B. Below are the match scores on NQ:
>
> - BGE Large 0.34B: 10.39
> - BGE Base 0.11B: 10.31
> - BGE Small 0.03B: 10.17
>
> From the paper:
> - GritLM 7B: 30.50
>
> We find performance to be significantly worse than with GritLM. Based on a manual inspection of samples, it appears that the embedding models commonly retrieve irrelevant passages that confuse the generative model. There may be other smaller embedding models or other generative models that may perform better, but overall we expect the RAG performance to be a function of the embedding and generative performance of the individual components (e.g. if an embedding model performs better than GritLM, we would expect it to lead to better RAG performance; BGE generally does not perform better on embedding as shown in Table 1). We have added this in Appendix F, thank you for raising it!
>
> **Modularity:** This is an interesting topic, thanks for bringing it up! One can only use the embedding/generative part of GritLM, thus it is still modular. However, in that case, some advantages to having the unified model are gone, such as e.g. query caching. The doc caching technique we introduce, however, still works even if embedding and generative models are separate. In that case, however, the entire corpus needs to be passed through the generative model once during index construction. From the compute perspective, retraining a GRIT model can be cheaper than a traditional RAG model. For GRIT, the pretraining and finetuning is both done using the same model, whereas for traditional RAG models, the embedding and generative model need to be pretrained and finetuned separately thus incurring more compute. We have added a note on this in the paper by rephrasing the end of the introduction, thanks for bringing it up!
>
> **More recent work:** Thank you for pointing us to these great works. We have added citations to them and several other recent embedding papers. Please let us know if there are any other works we should be citing.

---

> > ### Author Response · Authors · 2024-11-25
> > **Follow-up**
> >
> > Dear Reviewer,
> >
> > We'd appreciate it if you'd let us know if our response has addressed your concerns.
> >
> > Thanks!

---

> > > ### Comment · Reviewer_XQzj · 2024-11-25
> > > **Response to the authors**
> > >
> > > Thank you for your response. I appreciate the authors adding additional results in Appendix F. This is great! I also want to thank the authors for pointing out that GritLM is still modular.
> > >
> > > For these reasons, I will be increasing my score to 8 and confidence to 4.

---

### Author Response · Authors · 2024-11-19

We thank all reviewers for their detailed reviews and great feedback! Below is a summary of all changes we have made to the paper in a new uploaded revision:
1. Added RAG results with BGE in “Appendix F: Additional RAG results” in response to Reviewer XQzj.
2. Rephrased the end of the Introduction to better motivate that GritLM requires less compute than separate generative and embedding models when considering pretraining thanks to a pointer from Reviewer XQzj.
3. Added citations to more recent work in Section 3.2 thanks to pointers from Reviewer XQzj.
4. Added more discussion on the potential speed-performance trade-off of using a smaller and faster embedding model by using the embedding from intermediate layers of GritLM in “Appendix A: Ablations” together with our embedding head ablation that explores the cost-performance trade-off of smaller embedding dimensions.
5. Added resources used by Gen.-only and Emb.-only baselines in “Appendix P: Hardware” thanks to the comment by Reviewer 2rfu.
6. Elaborated more on training discussion and potential avenues for improvement in “Appendix Q: Limitations and Future Work” in response to Reviewer 2rfu.

Overall, we are glad reviewers have found the paper to be well-positioned and the methods to be original and novel. Reviewers have also pointed to the strong performance of the model and its usefulness. There was a lot of interest in using GRIT for RAG and the caching variants proposed - We are excited about further pushing these approaches together with the broader community.

---

### Meta-Review · Area_Chair_JB4U · 2024-12-21

**Metareview:**

Previous embedding models and generative models are typically learned separately. This paper proposes to learn them together through massive multi-task training and different tasks are separated through instructions. Experimental results are strong, demonstrating the performance of this joint model can match the best performance from both worlds. One unique advantage of this model is the improved efficiency in RAG applications, where the model can reuse the encodings of the query in an RAG pipeline. The reviewers unanimously vote by acceptance of this paper with scores 6,6,8,8.

**Additional Comments On Reviewer Discussion:**

The reviewers praised the GRIT framework for its originality, unified approach to generative and embedding tasks, strong performance on benchmarks, and efficiency in Retrieval-Augmented Generation (RAG). Key concerns included scalability, modularity, storage costs for caching, the dependency on instructions, and the lack of experiments with other base models or resource comparisons. The authors addressed these by providing additional results, clarifying GRIT's modularity and caching techniques, discussing storage optimizations, emphasizing the robustness of instructions at larger scales, and adding experiments with different base models and detailed resource usage comparisons. I feel these concerns are satisfactorily addressed by the authors.

---

### Decision · Program_Chairs · 2025-01-22

Accept (Poster)